# Attenuation of RNA polymerase II pausing mitigates BRCA1-associated R-loop accumulation and tumorigenesis

Xiaowen Zhang[1,*], Huai-Chin Chiang[1,*], Yao Wang[1,*], Chi Zhang[1], Sabrina Smith[1], Xiayan Zhao[1], Sreejith J. Nair[1], Joel Michalek[2], Ismail Jatoi[3], Meeghan Lautner[3], Boyce Oliver[3], Howard Wang[3], Anna Petit[4], Teresa Soler[4], Joan Brunet[5], Francesca Mateo[6], Miguel Angel Pujana[6], Elizabeth Poggi[7], Krysta Chaldekas[7], Claudine Isaacs[7], Beth N. Peshkin[7], Oscar Ochoa[8], Frederic Chedin[9], Constantine Theoharis[10], Lu-Zhe Sun[11], Tyler J. Curiel[12], Richard Elledge[12], Victor X. Jin[1], Yanfen Hu[1] & Rong Li[1]

Most *BRCA1*-associated breast tumours are basal-like yet originate from luminal progenitors. BRCA1 is best known for its functions in double-strand break repair and resolution of DNA replication stress. However, it is unclear whether loss of these ubiquitously important functions fully explains the cell lineage-specific tumorigenesis. *In vitro* studies implicate BRCA1 in elimination of R-loops, DNA-RNA hybrid structures involved in transcription and genetic instability. Here we show that R-loops accumulate preferentially in breast luminal epithelial cells, not in basal epithelial or stromal cells, of *BRCA1* mutation carriers. Furthermore, R-loops are enriched at the 5′ end of those genes with promoter-proximal RNA polymerase II (Pol II) pausing. Genetic ablation of *Cobra1*, which encodes a Pol II-pausing and BRCA1-binding protein, ameliorates R-loop accumulation and reduces tumorigenesis in *Brca1*-knockout mouse mammary epithelium. Our studies show that Pol II pausing is an important contributor to *BRCA1*-associated R-loop accumulation and breast cancer development.

[1] Department of Molecular Medicine, University of Texas Health Science Center at San Antonio, 8403 Floyd Curl Drive, San Antonio, Texas 78229, USA. [2] Department of Epidemiology and Biostatistics, University of Texas Health Science Center at San Antonio, San Antonio, Texas 78229, USA. [3] Department of Surgery, Cancer Therapy and Research Center, University of Texas Health Science Center at San Antonio, San Antonio, Texas 78229, USA. [4] Department of Pathology, University Hospital of Bellvitge, Bellvitge Institute for Biomedical Research (IDIBELL), L'Hospitalet del Llobregat, Barcelona 08908, Spain. [5] Hereditary Cancer Program, Catalan Institute of Oncology (ICO), Girona Biomedical Research Institute (IDIBGI), Girona 17007, Spain. [6] Breast Cancer and Systems Biology Lab, Program Against Cancer Therapeutic Resistance (ProCURE), Catalan Institute of Oncology (ICO), Bellvitge Institute for Biomedical Research (IDIBELL), L'Hospitalet del Llobregat, Barcelona 08908, Spain. [7] Lombardi Comprehensive Cancer Center, Georgetown University, Washington, District of Columbia 20007, USA. [8] PRMA Plastic Surgery, San Antonio, Texas 78240, USA. [9] Department of Molecular and Cellular Biology, University of California, Davis, California 95616, USA. [10] South Texas Pathology Associates, San Antonio, Texas 78229, USA. [11] Department of Cell Systems and Anatomy, University of Texas Health Science Center at San Antonio, San Antonio, Texas 78229, USA. [12] Department of Medicine, University of Texas Health Science Center at San Antonio, San Antonio, Texas 78229, USA. * These authors contributed equally to this work. Correspondence and requests for materials should be addressed to V.X.J. (email: jinv@uthscsa.edu) or to Y.H. (email: huy3@uthscsa.edu) or to R.L. (email: lir3@uthscsa.edu).

Germ-line *BRCA1* mutations are associated with significantly increased breast cancer incidence. Breast epithelium consists of two layers: an inner layer of luminal progenitors and mature luminal cells, and a basal layer consisting of mammary stem cells and differentiated myoepithelial cells[1]. Although most *BRCA1*-associated breast tumours are basal-like, they originate from luminal progenitor cells[2–4]. Unlike their counterparts from non-carriers, luminal progenitor cells from *BRCA1* mutation carriers exhibit hormone-independent proliferation[2] and attenuation of gene expression associated with luminal differentiation[4]. More recent studies indicate that the RANK–RANKL axis that drives paracrine actions in luminal homeostasis is aberrantly activated in *BRCA1* mutation carriers[5,6]. At the molecular level, BRCA1 is best known for its roles in supporting homologous recombination (HR)-based double-strand break repair[7–9] and suppressing DNA replication stress[10–15]. However, it is not known whether loss of these functions, which are ubiquitously important to all proliferating cells, is sufficient to account for cell lineage-specific tumorigenesis in breast epithelium of women carrying *BRCA1* mutations.

In addition to HR repair and DNA replication stress, BRCA1 is implicated in transcriptional regulation[7,16] and chromatin reorganization[17,18]. Recent cell-line studies indicate that BRCA1 also has a role in elimination of R-loops, transcriptional by-products that influence gene regulation and genomic integrity[19–21]. We recently found HR repair-independent functional antagonism between BRCA1 and cofactor of BRCA1 (COBRA1)[22] during mouse mammary gland development. COBRA1, also known as NELFB, is a BRCA1-binding protein and an integral subunit of the RNA polymerase II (Pol II)-pausing, negative elongation factor (NELF)[17,23,24]. Consistent with a functional role of BRCA1 in transcription, genome-wide studies found preferential association of BRCA1 with transcription start sites (TSS) in the human genome[25–27]. However, whether a role of BRCA1 in transcription directly contributes to BRCA1-associated tumorigenesis remains unclear.

Here we survey genome-wide R-loop dynamics in different breast cell types from *BRCA1* mutation carriers and non-carriers. We found that *BRCA1* mutation-associated R-loops preferentially accumulate in luminal epithelial cells and at genomic loci with paused Pol II. Using mouse genetic models, we further show that attenuation of Pol II pausing reduces incidence of *Brca1*-associated mouse mammary tumour incidence in a DNA repair-independent manner. Our work thus uncovers a previously unappreciated functional antagonism between BRCA1 and Pol II pausing in breast tumorigenesis.

## Results

**R-loops in breast tissue of *BRCA1* mutation carriers.** To ascertain clinical relevance of transcription-related BRCA1 functions, we first used immunofluorescence (IF) staining to compare R-loop intensity in formalin-fixed paraffin-embedded (FFPE), cancer-free breast tissue from *BRCA1* mutation-carrying women versus non-carriers. We found that R-loop intensity in *BRCA1* mutation carriers (B1, $n = 55$) was significantly higher than that in non-carriers (NC, $n = 36$, $P < 0.001$ by two-tailed *t*-test, Fig. 1a,b). Pretreatment of the FFPE samples with RNase H, which degrades RNA in R-loops, significantly reduced the IF staining in *BRCA1* mutation carriers (Fig. 1a), thus corroborating specificity of the IF signals. A cohort of *BRCA2* mutation carriers exhibited similar increase in R-loop intensity as compared to non-carriers, but the difference did not reach statistical significance (Supplementary Fig. 1). Notably, the vast majority of luminal epithelial cells in a typical *BRCA1* mutation

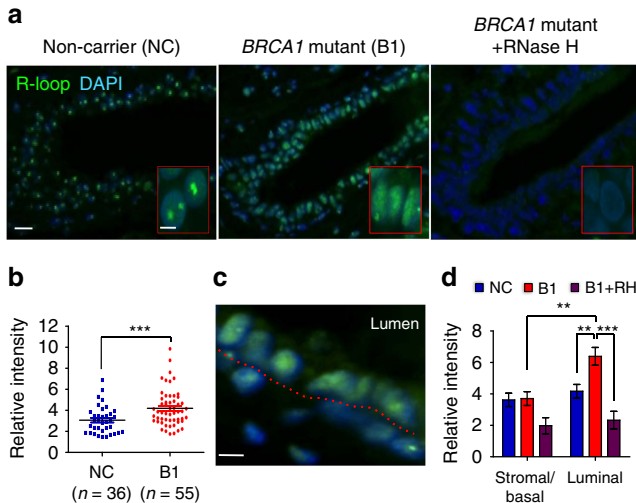

**Figure 1 | *BRCA1* mutation-associated R-loop accumulates preferentially in luminal breast epithelial cells.** (**a**) Low and high (inlet) magnification images of R-loop staining in samples from non-carriers and *BRCA1* mutation carriers, with and without pretreatment of RNase H. Scale bar, 20 µm (left) and 5 µm (right). (**b**) Quantitation of relative R-loop intensity in the non-carrier group (NC; $n = 36$) and *BRCA1* mutation carrier group (B1; $n = 55$). ***$P < 0.001$ by two-tailed *t*-test. (**c**) Image of R-loop staining in *BRCA1* mutation carriers. The dash line indicates the boundary between luminal epithelial cells and stromal–basal cells. Scale bar, 5 µm. (**d**) Quantitation of relative R-loop intensity in stromal–basal compartment and luminal epithelial compartment in the non-carrier group (NC; $n = 12$), *BRCA1* mutation carrier group (B1; $n = 12$) and *BRCA1* mutation carrier pretreated with RNase H group (B1 + RH; $n = 5$). **$P < 0.01$ and ***$P < 0.001$ by two-tailed *t*-test. Error bars represent s.e.m.

carrier sample exhibited elevated R-loop staining, whereas basal epithelial and stromal cells from the same *BRCA1* mutant specimen did not display higher intensity than their counterparts in non-carriers (Fig. 1c,d). This luminal cell-specific R-loop accumulation is reminiscent of the lineage-specific cell of origin for *BRCA1*-associated breast tumours.

**Genome-wide survey of R-loop accumulation.** To corroborate IF results and to identify the genomic locations of *BRCA1* mutation-associated R-loop accumulation, we sorted breast cells from fresh breast tissue and used them for R-loop-specific DNA-RNA immunoprecipitation-sequencing (DRIP-seq)[28]. Tissue samples from four *BRCA1* mutation carriers (B1) and four non-carriers (NC) were procured, digested into single cells and sorted by flow cytometry using established cell surface markers (EpCAM and CD49f; Fig. 2a and Supplementary Fig. 2)[29]. Four distinct cell populations were acquired: stromal cells, basal epithelial cells, luminal progenitor (LP) cells and mature luminal epithelial (ML) cells. Each sorted cell population was subjected to DRIP using an established protocol and R-loop-specific antibody[28]. DNA samples from DRIP reactions were amplified and used in deep sequencing. For bioinformatics analysis, extracted reads were normalized to total reads of a given sample (see Methods for details).

Using DRIP-seq data from the four *BRCA1* mutation carriers and four non-carriers, we found that R-loop levels in the two luminal cell populations (LP and ML) were more pronounced than basal epithelial and stromal cells from the same cohorts (compare columns 1–4 with 5–8 in Fig. 2b), indicating cell-type-specific R-loop accumulation regardless of the *BRCA1*

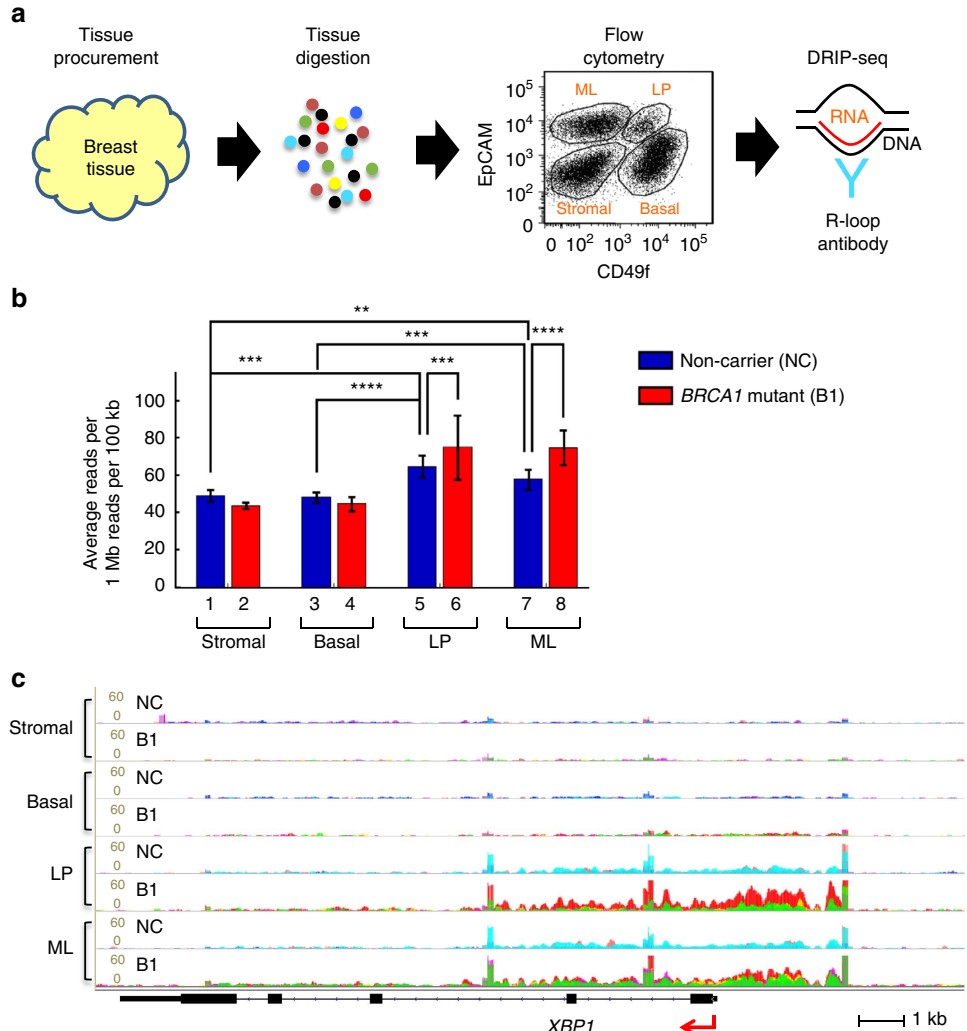

**Figure 2 | DRIP-seq validation of luminal lineage-specific R-loop accumulation in *BRCA1* mutation carriers.** (**a**) Experimental design for DRIP-seq. Fresh cancer-free breast tissue samples were digested into single cells, and then sorted by flow cytometry into four populations: stromal, basal, LP and ML. Genomic DNA from each population was extracted, digested and immunoprecipitated with an R-loop-specific antibody S9.6. DRIP DNA was subjected to deep sequencing. (**b**) Average reads per 1 Mb reads per 100 kb (from TSS to TTS) in stromal–basal–LP–ML population in non-carriers ($n = 4$) and *BRCA1* mutation carriers ($n = 4$). **$P \leq 0.01$, ***$P \leq 0.001$ and ****$P < 0.0001$ by permutation test. Error bars represent s.e.m. (**c**) Track view of DRIP-seq density profile centred on gene *XBP1*. Each track is an overlay of four individual non-carriers (NC) or four *BRCA1* mutation carriers (B1) indicated by different colours. TSS was marked by red arrow.

mutant status. Furthermore, LP and ML populations of *BRCA1* mutation carriers exhibited even higher R-loop levels than their counterparts from non-carriers (compare column 5 with 6, and 7 with 8 in Fig. 2b). In contrast, overall R-loop intensity in stromal and basal cell compartments was comparable between *BRCA1* mutation carriers and non-carriers (compare column 1 with 2, and 3 with 4). We confirmed the *BRCA1* mutation-associated R-loop enrichment by locus-specific PCR (Supplementary Fig. 3). Furthermore, pretreatment with RNase H before DRIP completely abolished the R-loop signals in locus-specific DRIP and genome-wide DRIP-seq (Supplementary Fig. 4a,b), thus corroborating the antibody specificity. Of note, a number of genes involved in luminal fate determination and differentiation showed luminal cell-specific enrichment of R-loop signals in *BRCA1* mutant samples, including *XBP1*, *GATA3*, *CEBPB* and *FOXC1* (multicolour overlay representing biological repeats in Fig. 2c and Supplementary Figs 5 and 8). Interestingly, the R-loop distribution at the *XBP1* locus *in vivo* bears striking resemblance to that of oestrogen-stimulated R-loops observed in a recent cell

culture study[30]. Taken together, DRIP-seq and IF using independent cohorts of fresh and FFPE clinical samples, respectively, lead us to the same conclusion that *BRCA1* mutation carriers are associated with luminal cell-specific R-loop accumulation in cancer-free breast tissue.

**Genomic features of R-loop accumulation.** Genome-wide analyses indicate that overall R-loop intensity in both *BRCA1* mutation carriers and non-carriers is most pronounced at the 5′ end of genes, followed by the 3′ end (Fig. 3a,b). As expected, the gene body, gene distal and gene desert regions had relatively low R-loop signals in both clinical cohorts. This finding is in line with a recent report of prevalent and conserved R-loop formation at promoter and terminator regions of Pol II-dependent genes in human and mouse genomes[31]. Consistent with our prior data, *BRCA1* mutation-associated R-loop elevation was observed at these genic hotspots only in cells of luminal lineage, but not in stromal or basal epithelial cells (Fig. 3a). We define genes with

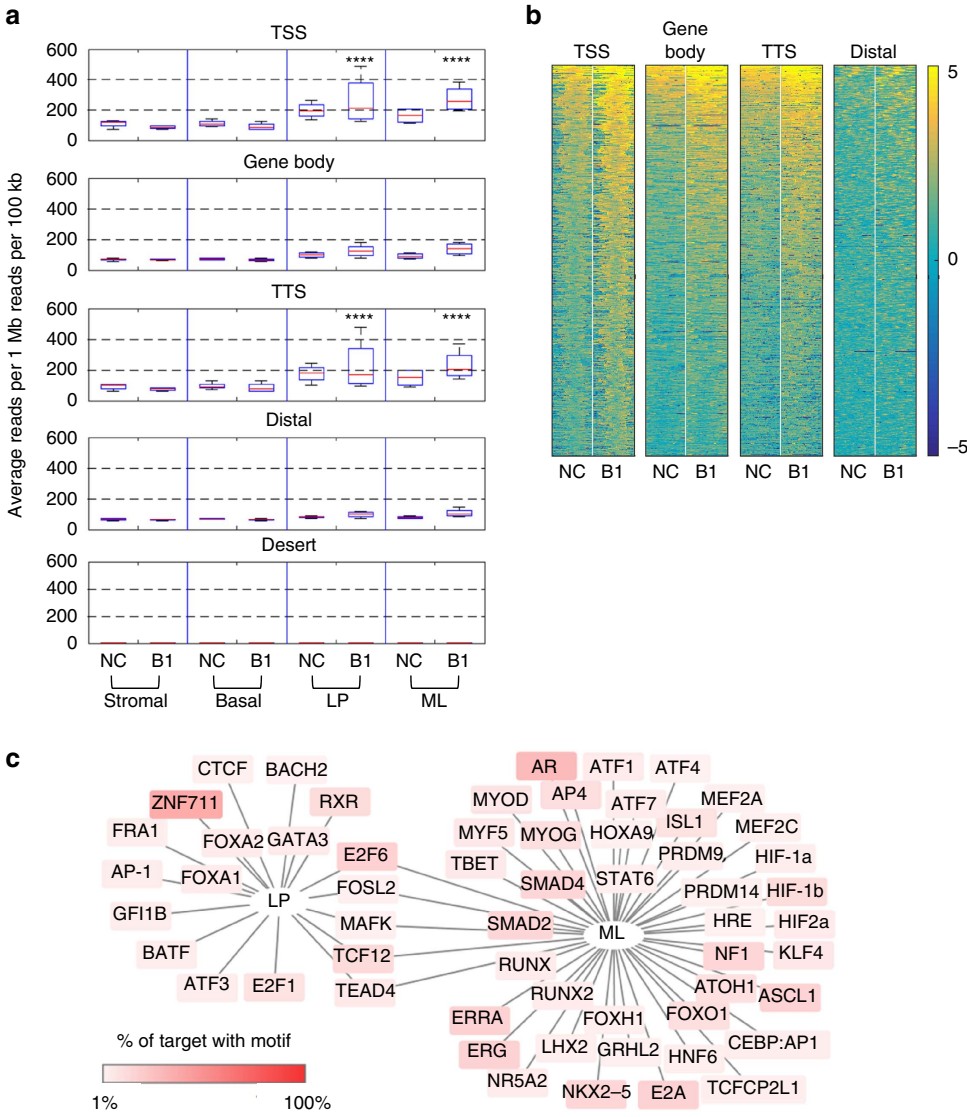

**Figure 3 | *BRCA1* mutation-associated R-loop accumulates preferentially at transcription regulatory regions.** (**a**) Box plot displaying average reads per 1 Mb reads per 100 kb in the peak regions in stromal–basal–LP–ML in non-carriers (NC; $n = 4$) and *BRCA1* mutation carriers (B1; $n = 4$). Peaks were identified using uniquely mapped reads, and then extended to the closest restriction enzyme cutting site. Extended peaks from all samples were combined, and categorized into TSS, gene body, TTS and distal peaks. For gene desert, reads from ∼11,000 gene desert regions, each with the length of 20 kb were used. Average reads per 1 Mb reads per 100 kb were calculated in each category. ****$P \leq 0.0001$ permutation test. (**b**) Heatmap of averaged DRIP-seq density in non-carriers (NC; $n = 4$) and *BRCA1* mutation carriers (B1; $n = 4$) centred on TSS, gene body, TTS and distal regions. Ranked by *BRCA1* mutation carriers high to low. Heatmaps were drawn using genes that are top-ranked 5,000 in TSS region. Genes in each subpanel of heatmap are sorted individually. (**c**) Cytoscape depicting HOMER motif analysis. Genes with *BRCA1* mutation-associated R-loop at TSS in either LP or ML population were used as target sequences for motif search. Background colour of motifs indicates the percentage of target sequences that has corresponding motif.

*BRCA1* mutation-associated R-loop at TSS as those in either of the following two groups: (1) TSS with R-loop peaks in both B1 and NC samples but average reads of B1 in TSS upstream/downstream 2 kb region is greater than NC with $\log_2 FC \geq 0.8$, adjusted $P$ value $\leq 0.05$; (2) TSS with common R-loop peaks only present in B1. Using a combined gene set consisting of both groups of genes, we found that promoter regions with *BRCA1* mutation-associated R-loops are enriched with binding sites for breast cancer-related transcription factors, such as GATA3 and FOXA1 in LP cells and SMAD2/4 and STAT6 in ML cells (Fig. 3c). There are also significant overlaps between our gene list and previously defined luminal signature genes (Supplementary Data 1). Furthermore, gene ontology indicates that mammary neoplasm is the top disease-associated category among genes with *BRCA1* mutation-associated R-loops (Supplementary Fig. 6).

**R-loop accumulation is associated with Pol II pausing**. Among those genes with common promoter-associated R-loops in *BRCA1* mutation carriers, 88% do not share R-loops at the corresponding terminator regions (Fig. 4a), suggesting that distinct mechanisms are involved in the regulation of R-loop dynamics at these two genic hotspots. In a recent genome-wide study of R-loops in the K562 human erythroleukemia cell line[31], Chedin and co-workers[31] showed that genes with promoter-proximal R-loop formation display a significantly higher level of nascent transcription immediately downstream of TSS, coinciding precisely with the hotspot of R-loop formation. This raised the distinct possibility that R-loop accumulation at TSS could be associated with promoter-proximal Pol II pausing. Using total Pol II chromatin immunoprecipitation-seq (ChIP-seq) data from the same published study, we calculated

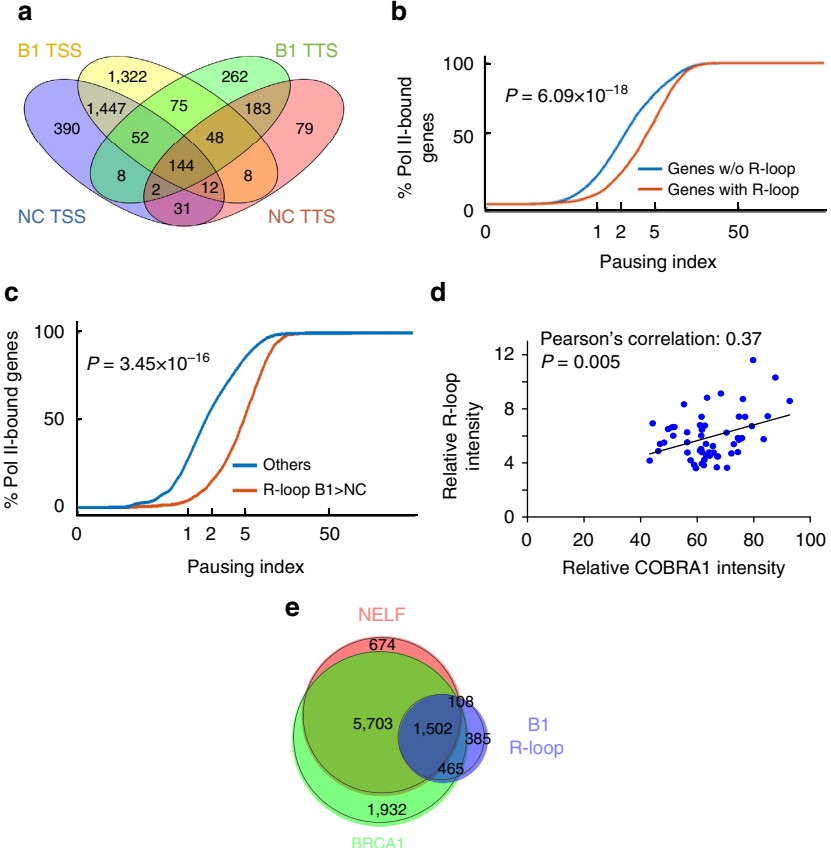

**Figure 4 | Association of Pol II pausing and *BRCA1* mutation-related R-loop accumulation.** (**a**) Venn diagram depicting genes with common TSS or TTS peaks in ML of non-carriers (NC; $n = 4$) and *BRCA1* mutation carriers (B1; $n = 4$). (**b**) Cumulative curve of pausing index for genes with TSS R-loop peaks (red) or without TSS R-loop peaks (blue) in K562 cells. *P* value was calculated using signed rank-sum test. (**c**) Cumulative curve of pausing index for genes with *BRCA1* mutation-associated TSS R-loops (red) and the rest of the reference genes (blue) in primary mammary epithelial cells. *P* value was calculated using signed rank-sum test. (**d**) Positive correlation between human COBRA1 immunohistochemistry and R-loop staining among the 55 *BRCA1* mutation carriers. $P = 0.005$. Pearson's correlation $= 0.37$. (**e**) Venn diagram depicting the overlap between genes with TSS-bound NELF, TSS-bound BRCA1 and genes with *BRCA1* mutation-associated TSS R-loops.

Pol II pausing index based on previously established definition (Supplementary Fig. 7a)[32]. In support, we found a strong positive correlation between genes with promoter-proximal R-loop peaks and Pol II-pausing index ($P = 6.09 \times 10^{-18}$ by signed rank-sum test, Fig. 4b). A similar correlation was found between Pol II pausing and high GC skew, a known predictor for promoter-proximal R-loop accumulation[28] ($P = 2.32 \times 10^{-15}$ by signed rank-sum test, Supplementary Fig. 7c). To corroborate these cell culture-based findings, we conducted total Pol II ChIP-seq using primary human breast epithelial cells from fresh human breast tissue. Again, we observed a remarkable concordance between genes with *BRCA1* mutation-associated R-loop and Pol II-pausing index at TSS *in vivo* ($P = 3.45 \times 10^{-16}$ by signed rank-sum test, Fig. 4c). Taken together, these correlative data are consistent with the notion that promoter-proximal Pol II pausing and R-loop accumulation are mechanistically linked.

**COBRA1 depletion ameliorates BRCA1-associated R-loops.** The four-subunit NELF complex plays a key role in Pol II pausing[33]. We previously reported a physical association between BRCA1 and one of the NELF subunits, NELFB/COBRA1 (ref. 17). In a more recent study, we provided compelling mouse genetic evidence that BRCA1 and COBRA1 functionally antagonize each other during normal mammary gland development[22]. We

therefore postulated that the same functional antagonism could extend to the regulation of R-loop dynamics in such a manner that NELF-mediated Pol II pausing leads to elevated R-loop signals in BRCA1-deficient cells. Consistent with this notion, we found that the expression of human COBRA1 positively correlates with R-loop intensity in breast tissue of *BRCA1* mutation carriers ($P = 0.005$, Pearson's correlation 0.37, Fig. 4d). Furthermore, using publicly available ChIP-seq data for BRCA1 and NELF[26,34,35], we observed a significant overlap between BRCA1- and NELF-bound genes ($P = 0$ by Fisher's test, Fig. 4e). An extensive overlap was also found between BRCA1/NELF-bound genes and those with elevated TSS R-loop signal in *BRCA1* mutation carriers ($P = 0$ by Fisher's test, Fig. 4e).

To explore a functional link between promoter-proximal NELF-mediated Pol II pausing and *BRCA1*-associated R-loop accumulation, we knocked down (KD) BRCA1 and COBRA1 in T47D luminal breast cancer cells (Fig. 5a). Locus-specific DRIP showed that BRCA1 KD increased R-loop levels at two genic regions that displayed elevated R-loop enrichment *in vivo* (compare column 1 with 2 in Fig. 5b,c; Supplementary Fig. 8). This effect of BRCA1 KD was verified by multiple independent BRCA1-targeting short interfering RNA (siRNA) oligos (Supplementary Fig. 9). Furthermore, the R-loop signals at both loci were sensitive to RNase H pretreatment of genomic DNA (columns 3–4 in Fig. 5b,c), thus confirming specificity of the DRIP assay. As expected, depletion of COBRA1 significantly

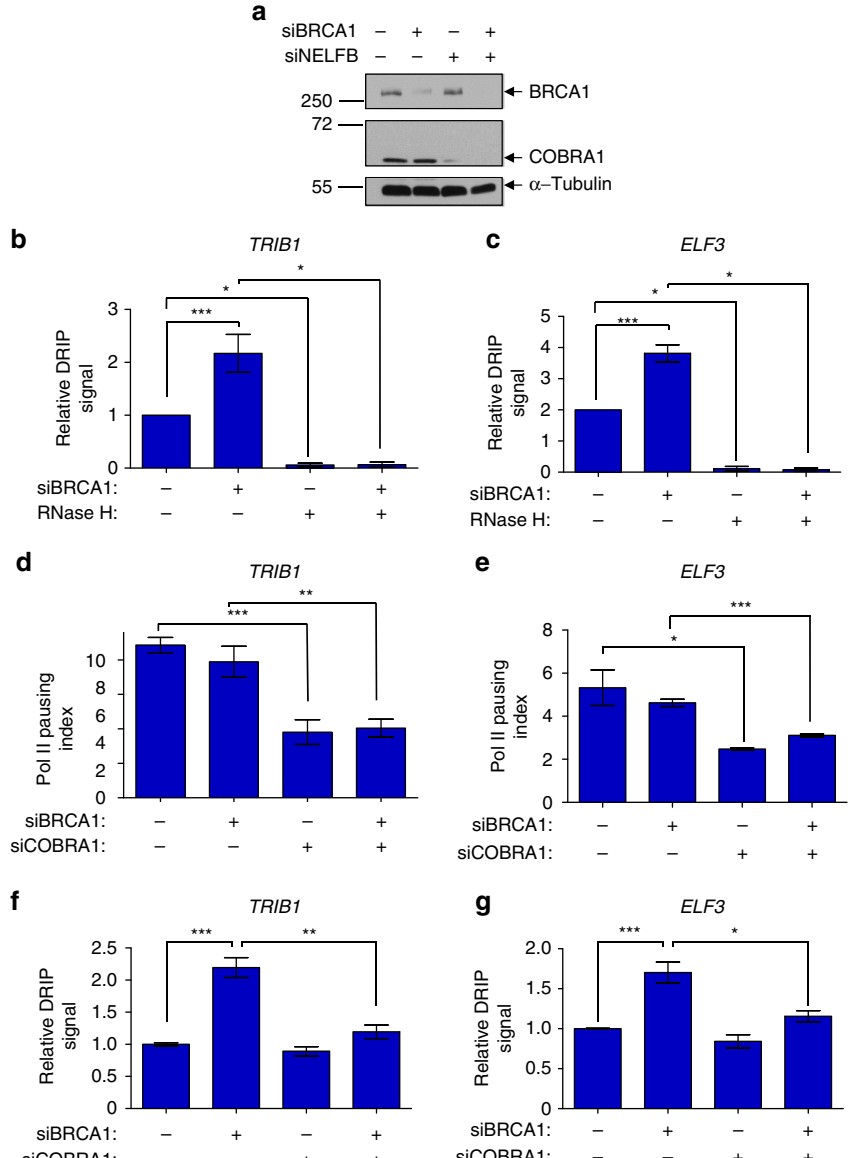

**Figure 5 | Depletion of COBRA1 ameliorates *BRCA1* mutation-related R-loop accumulation in T47D breast cancer cells.** (**a**) Immunoblots of BRCA1 and COBRA1 in knockdown T47D cells. α-Tubulin was used as the loading control. (**b** and **c**) Relative DRIP signal at (**b**) *TRIB1* and (**c**) *ELF3* loci in control and BRCA1 KD cells, with or without treatment of RNase H. The graph is an average of three individual experiments. (**d** and **e**) Pol II pausing index at (**d**) *TRIB1* and (**e**) *ELF3* loci in BRCA1 and/or NELFB KD T47D cells. Pol II pausing index is calculated by the ratio of total Pol II signals at TSS over gene body. The graph is an average of at least four individual experiments. (**f** and **g**) Relative DRIP signal at (**f**) *TRIB1* and (**g**) *ELF3* loci in BRCA1 and/or COBRA1 KD T47D cells. The graph is an average of at least six individual experiments. *P < 0.05, **P < 0.01 and ***P < 0.001 by two-tailed *t*-test. Error bars represent s.e.m.

reduced Pol II-pausing index in both control and BRCA1 KD cells (compare columns 1–2 with 3–4 in Fig. 5d,e), as calculated based on the ratio of total Pol II ChIP signal at TSS over that at a region in the gene body (Supplementary Fig. 7a,b). Of note, BRCA1 KD alone did not affect Pol II pausing (compare column 1 with 2 in Fig. 5d,e). We interpret this to mean that BRCA1 acts downstream of NELF-dependent Pol II pausing to alleviate pausing-induced R-loop accumulation. Consistent with this notion, ectopic expression of human RNase H1 in T47D cells reduced R-loop intensity but not promoter-proximal Pol II pausing (Supplementary Fig. 10). In further support of our hypothesis, COBRA1 KD resulted in a significant reduction in R-loop levels in BRCA1 KD cells (compare columns 2 with 4 in Fig. 5f,g). Interestingly, COBRA1 KD did not lead to any appreciable decrease in basal R-loop levels in control cells (compare columns 1 with 3 in Fig. 5f,g). One explanation is that

the R-loop-attenuating activity of BRCA1 in control cells is potent enough to prevent R-loop accumulation caused by NELF-dependent Pol II pausing.

***Cobra1* ablation mitigates mammary tumorigenesis.** To investigate the functional interaction between COBRA1 and BRCA1-associated R-loop dynamics *in vivo*, we assessed R-loop intensity in mammary gland-specific (*MMTV-Cre*) KO mice for *Brca1* (BKO), *Cobra1* (CKO) and both genes (double knockout (DKO))[22]. Consistent with our IF findings from human *BRCA1* mutation carriers, the vast majority of luminal cells in BKO mammary glands exhibited significantly higher R-loop staining than their wild-type (WT) counterparts (Fig. 6a,b). The IF signal was significantly diminished by pretreatment with RNase H (+RH, Fig. 6a,b), confirming antibody specificity. Consistent

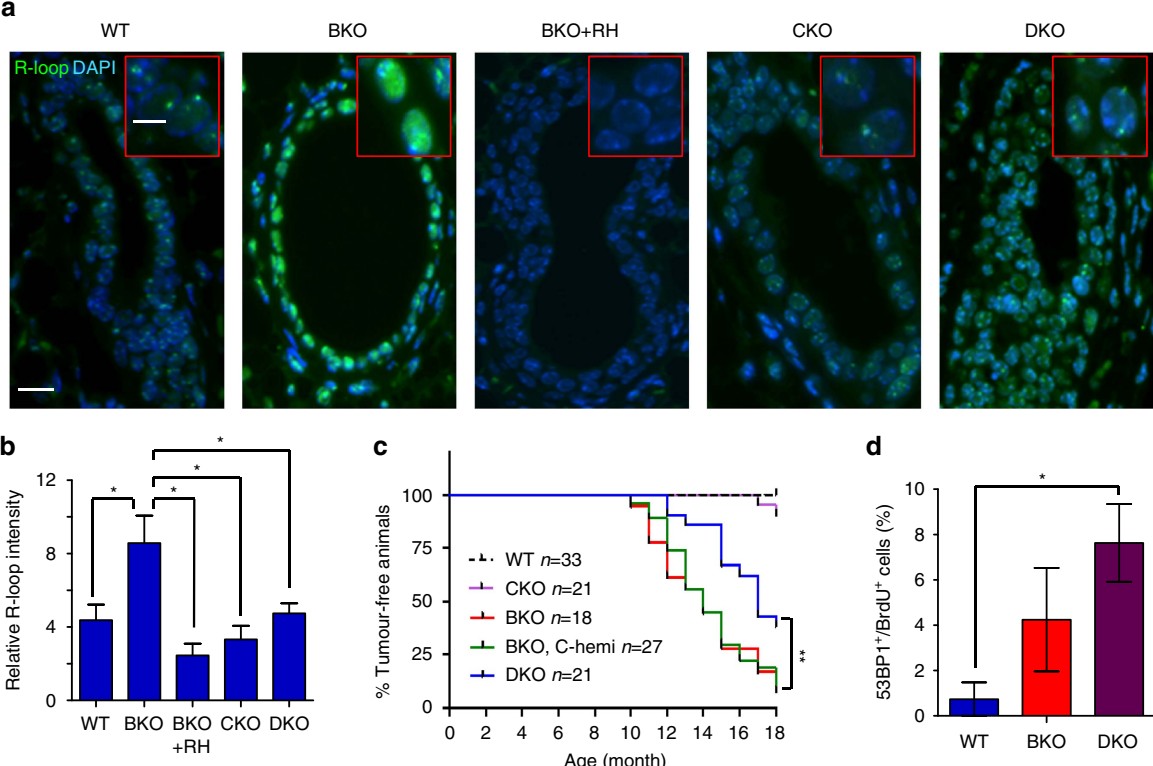

**Figure 6 | Ablating COBRA1 mitigates BRCA1-associated R-loop accumulation and mammary tumorigenesis.** (**a**) Low and high (inlet) magnification images of R-loop staining in mammary ducts of 8-week-old virgin mice, with and without pretreatment of RNase H (RH). Scale bars, 5 μm (top) and 20 μm (bottom). (**b**) Quantitation of relative R-loop intensity in 8-week-old animals. The number of animals used in each group is: WT ($n = 9$), BKO ($n = 9$), BKO + RH ($n = 3$), CKO ($n = 5$) and DKO ($n = 8$). *$P < 0.05$ by two-tailed $t$-test. Error bars represent s.e.m. (**c**) Kaplan–Meier curve for tumour incidence. **$P < 0.01$ by log-rank test. (**d**) Percentage of BrdU$^+$ mouse mammary epithelial cells with 53BP1 foci from 12-week-old mice. The number of animals used in each group is: WT (3), BKO (3) and DKO (3). *$P < 0.05$ by two-tailed $t$-test. Error bars represent s.e.m.

with the findings of COBRA1 KD in cell culture, *Cobra1* ablation alone in mammary epithelium *in vivo* did not exhibit statistically significant difference in R-loop intensity compared to their WT counterparts (Fig. 6b). Remarkably, mammary epithelial cells of *Brca1/Cobra1* DKO displayed significantly reduced R-loop levels versus those in BKO (Fig. 6a,b). This suggests that the R-loop accumulation observed in *Brca1*-deleted mammary epithelium is, to a large part, caused or reinforced by the action of COBRA1.

We then asked whether COBRA1 affects *Brca1*-associated mouse mammary tumorigenesis. Consistent with published findings[36], BKO mice had increased spontaneous mammary tumours (red, Fig. 6c). Hemizygous deletion of *Cobra1* in the BKO background (BKO, C-hemi) did not affect *Brca1*-associated mammary tumorigenesis (green). In striking contrast, DKO mice with homozygous deletion of *Cobra1* and *Brca1* (blue) exhibited significantly lower mammary tumour incidence than BKO and BKO, C-hemi mice (Fig. 6c). Despite the significant difference in tumour incidence, tumours from DKO and BKO displayed similar degrees of cell proliferation and apoptosis as measured by Ki67 staining and TdT-mediated dUTP nick end labelling assay, respectively (Supplementary Fig. 11). Thus, genetic ablation of *Cobra1* ameliorates *Brca1*-associated R-loop accumulation and mammary tumour development, but it does not appear to affect progression of *Brca1*-associated tumours once they are formed.

BRCA1 functions in HR repair[9] and suppression of DNA replication stress[10–15] have been well documented. Furthermore, a recent study shows that oestrogen-dependent R-loop formation is associated with DNA replication-dependent double-strand breaks[30]. Thus, *Cobra1* ablation could reduce *Brca1*-associated tumorigenesis by restoring HR repair and/or reducing elevated

DNA replication stress in *Brca1*-deficient mammary epithelial cells. However, using both cell culture-based assay and mouse genetic models, our published studies showed that mammary epithelial cells of DKO mice exhibited the same degree of HR repair deficiency as those in BKO[22]. Furthermore, DNA replication stress as measured by 53BP1 staining among 5-bromodeoxyuridine-positive (BrdU$^+$) cells was even more pronounced in DKO versus BKO mammary epithelium (Fig. 6d). Thus, reduced tumour incidence in DKO mice is unlikely due to rescue of either HR repair defect or elevated DNA replication stress associated with BRCA1 deficiency.

## Discussion

Combined studies of clinical samples, cell culture and genetically engineered mouse models lead us to the conclusion that NELF-mediated Pol II pausing is an important source of R-loop accumulation in BRCA1-deficient mammary epithelial cells. Importantly, attenuation of Pol II pausing can mitigate R-loop accumulation and reduce incidence of mammary tumour development due to BRCA1 deficiency. It is conceivable that nascent transcripts produced by promoter-proximally paused Pol II are prone to R-loop formation. We propose that, through its direct association with COBRA1 and perhaps other transcription factors, BRCA1 is brought to genes with NELF-dependent Pol II pausing to ameliorate promoter-proximal R-loop accumulation. Reduced BRCA1 expression and/or activity, while not affecting NELF-dependent Pol II pausing *per se*, can lead to excessive elevation of R-loop signals at those gene loci. Although NELF-dependent Pol II pausing is important for development-

related transcription[22], the ensuing R-loop accumulation in the absence of BRCA1 could ultimately contribute to tumorigenesis. Thus, functional antagonism between BRCA1 and NELF enforces a fine balance between normal tissue development and tumour suppression.

Attenuated R-loop levels in DKO mice clearly correlates with reduced tumour incidence. Consistent with previous reports, we found that BRCA1-deficient mouse mammary epithelium is associated with compromised HR repair[22], elevated DNA replication stress (Fig. 6d) and increased RANKL expression (Supplementary Fig. 12a). However, unlike R-loop levels and tumour incidence, none of these abnormalities in BKO mice was rescued on further deletion of *Cobra1* in DKO mice, nor were there any significant changes in levels of oestrogen receptor or progesterone receptor between BKO and DKO (Supplementary Fig. 12b). Compared to their corresponding WT littermates, neither BKO nor DKO exhibited substantial transcriptomics changes in mammary epithelia (Supplementary Fig. 13 and Supplementary Data 2). Future studies will shed light on the underlying mechanism by which BRCA1/COBRA1-regulated R-loop dynamics influences BRCA1-associated tumour development.

DRIP-seq of clinical samples from both *BRCA1* mutation carriers and non-carriers identifies significantly elevated overall R-loop levels in luminal cells compared to basal and stromal cells. A recent epigenomic and transcriptomic study of human breast showed that luminal cells have twice the number of hypomethylated transcription enhancers and approximately four times the amount of total RNA compared with myoepithelial cells[37]. This remarkable asymmetry of epigenetics status and steady-state transcript abundance is consistent with the notion that the higher R-loop levels in luminal cells result from more active nascent transcription. However, direct measurement of nascent transcripts in human breast is technically challenging, because of the extensive tissue processing *in vitro* and limited number of resulting sorted primary cells. A related question concerns the luminal-specific R-loop accumulation in *BRCA1* mutation carriers. Because no R-loop elevation was observed at transcriptionally active genes in either stromal or basal cells of the same *BRCA1* mutation-carrying individuals, the *BRCA1* mutation-associated R-loop elevation in luminal cells is unlikely to be a mere by-product of active transcription *per se*. BRCA1 is functionally and/or physically associated with a number of luminal cell-specific transcription factors that are involved in fate determination, including ERα, GATA3 and FOXA1 (refs 7,16,38,39). These functional links could in turn confine BRCA1 action to transcriptional regulatory regions in the luminal cell compartment.

MMTV-Cre-based gene KO tends to be biased towards the luminal compartment of mouse mammary epithelium. Nevertheless, BRCA1 expression in BKO mice used in the current study was reduced equally effectively in luminal and basal epithelial compartments (Supplementary Fig. 14). Thus, data from both mouse models and human clinical samples are valid in demonstrating cell lineage-specific R-loop enrichment. However, we realize that *Brca1* KO mouse models do not entirely recapitulate disease development in human *BRCA1* mutation carriers. For example, unlike humans, tumour incidence is not higher in heterozygous *Brca1* KO mice versus WT mice[40]. Despite these species-dependent differences, parallel investigation of both clinical samples and mouse genetic models promises to shed more light on a clinically relevant and biologically underinvestigated problem.

Genetic complementation between mouse *Brca1* and *Cobra1* in R-loop dynamics and mammary tumorigenesis strongly suggests that COBRA1 contributes to BRCA1-associated breast cancer

development. This is further supported by the positive correlation between COBRA1 expression and R-loop intensity in breast epithelium of *BRCA1* mutation carriers. It is also worth noting that tumour samples from triple-negative breast cancer patients, which contain the majority of *BRCA1* mutation carriers, had significantly elevated COBRA1 mRNA as compared to normal breast epithelium from cancer-free individuals[41] (2.55-fold, $P = 4.83 \times 10^{-6}$ by two-tailed *t*-test). In summary, our study suggests that NELF-dependent Pol II pausing is a previously unappreciated contributor to tumour development in *BRCA1*-deficient breast epithelium, which offers a potential cancer risk-assessing and -reducing tool.

## Methods

**Mice.** *Cobra1*[f/f] mice have been described previously[42]. *Brca1*[f/f] mice[36] were obtained from Mouse Model of Human Cancer Consortium, National Cancer Institute. *MMTV-Cre* line A mice (from Dr Anthony Wynshaw-Boris) were used to generate *MMTV-Cre,Brca1*[f/f], *MMTV-Cre,Cobra1*[f/f] and *MMTV-Cre,Brca1*[f/f], *Cobra1*[f/f] as described previously[22]. Mutant mice and their littermate controls in the same mixed genetic background (129SvEv/SvJae/C57BL6/FVB) were analysed. All procedures performed on animals were approved by the Institutional Animal Care and Use Committee at the University of Texas Health Science Center at San Antonio.

**Breast tissue cohorts.** Cancer-free breast tissue was procured from women either undergoing cosmetic reduction mammoplasty, diagnostic biopsies or mastectomy, following protocols approved by the Institutional Review Board at the Georgetown University, the Catalan Institute of Oncology, PRMA Plastic Surgery and the University of Texas Health Science Center at San Antonio. All donors signed written consent forms authorizing the use of the specimens for breast cancer-related laboratory investigations.

**Cell culture.** T47D was purchased from ATCC and cultured in high glucose DMEM (Thermo Fisher Scientific; 11965) supplemented with 10% foetal bovine serum (FBS), 100 μg ml$^{-1}$ penicillin and 100 μg ml$^{-1}$ streptomycin (Thermo Fisher Scientific; 15140122). siRNA pools for non-targeting control (Dharmacon; D-001810-10) was purchased. Individual siRNAs were synthesized from Sigma-Aldrich. siRNA target sequences are listed in Supplementary Table 1. siRNA transfection was carried out using Lipofectamine RNAiMAX (Thermo Fisher Scientific; 13778150) following the manufacturer's instruction. Briefly, 20 nM of siRNA was transfected with 25 μl of RNAiMAX reagent. Cells were collected 3 days after siRNA transfection.

Human *RNASEH1* gene (Addgene; 65782) was subcloned into pCDH-EF1-MCS-T2A-Puro lentivector (System Biosciences; CD520A-1) via *Eco*RI/*Bam*HI restriction sites. Control and RNase H1 lentiviruses were produced in HEK293T cells by co-transfecting cells with the lentiviral vectors and the corresponding helper plasmids using Lipofectamine 2000 (Thermo Fisher; 11668019). Viral supernatant was collected 48 h after transfection, and was passed through a 0.45 μM filter before titration. Lentiviruses were titred using a Quantitative PCR-Based Lentivirus Titration Kit (Applied Biological Materials; LV900) following the manufacturer's instruction. Control and RNase H1 lentiviruses of the same titre were used to infect T47D cells with 0.8 μl ml$^{-1}$ polybrene (Millipore; TR-1003-G). T47D cells were collected 72 h after infection and proceeded to DRIP and ChIP assays. Immunoblotting was performed to confirm RNase H1 overexpression (Proteintech; 15606-1-AP). PCR with reverse transcription for mRNA abundance was carried out using the primers listed in Supplementary Table 2.

**Protein IF/immunohistochemistry staining.** Twelve-week-old virgin mice were intraperitoneally injected with cell proliferation labelling reagent (GE Healthcare; RPN201) at 16.7 ml kg$^{-1}$. After 3 h, their inguinal mammary glands were removed and fixed with 10% neutral-buffered formalin (Fisher Scientific; 23245685) at 4 °C overnight. Fixed glands were paraffin embedded and cut into 3 μM sections for staining. Slides were baked at 70 °C for 15 min, and then deparaffinized/rehydrated by 100% xylene, 100% xylene, 100% xylene, 100% ethanol, 100% ethanol, 95% ethanol, 70% ethanol, 50% ethanol for 3 min each. After briefly washing with PBS, slides were boiled with antigen-unmasking solution (Vector Labs; H-3300) for 20 min. Slides were cooled down to room temperature, washed with PBS and blocked with 10% normal goat serum in PBS at room temperature for 1 h. Primary antibody incubation was done at 4 °C overnight. Primary antibodies used were: anti-BrdU (GE Healthcare; RPN20; 1:10,000), anti-53BP1 (Bethyl Laboratories; A300-272A; 1:500). After primary antibody incubation, slides were washed with PBS and incubated with Alexa-488- and Alexa-546-conjugated secondary antibodies (Life Technologies) at room temperature for 2 h. Slides were washed and mounted with Vectashield mounting medium with 4′,6-diamidino-2-phenylindole (DAPI) (Vector Labs; H-1200). A DeadEnd Fluorometric TUNEL System Kit

(Promega; G3250) was used for *in situ* nick-end labelling of DNA fragmentation according to the manufacturer's instructions.

For immunohistochemistry staining, slides were deparaffinized/rehydrated, boiled with antigen-unmasking solution, cooled down and washed as mentioned above. Slides were pretreated with 3% hydrogen peroxide for 10 min before blocking. Primary antibodies used were: anti-COBRA1 as described previously[24], anti-Ki67 (Thermo Fisher Scientific; MA5-14520; 1:100) and anti-RANKL (R&D Systems; AF462; 1:300). For detection of primary antibody using immune enzymatic method, the ABC Peroxidase Detection System (Vector Labs; PK-6105) was used with 3, 3′-diaminobenzidine (DAB) as substrate (Vector Labs; SK-4105) according to the manufacturer's instruction.

**R-loop IF staining.** Freshly cut 3 µM FFPE samples were baked at 70 °C for 15 min. After deparaffin and rehydration, samples were treated with boiling antigen-unmasking solution for 1 h. Samples were cooled down to room temperature, and then treated with $0.2 \times$ SSC buffer (Ambion; AM9763) with gentle shaking at room temperature for 20 min. Primary antibody incubation was done with monoclonal antibody S9.6 (Karafast; ENH001) at 1:100 dilution in PBS containing 1% normal goat serum and 0.5% Tween-20 at 37 °C overnight. After primary antibody incubation, samples were washed three times with PBS containing 0.5% Tween-20, and then incubated with Alexa-488-conjugated secondary antibody at 1:1,000 dilution in PBS containing 1% normal goat serum and 0.5% Tween-20 at 37 °C for 2 h. Samples were washed two times with PBS containing 0.5% Tween-20, two times with PBS, and then mounted with Vectashield mounting medium with DAPI. For samples pretreated with RNase H, an overnight treatment of RNase H (NEB; M0297S) was carried out after $0.2 \times$ SSC treatment. Samples were washed three times with PBS before primary antibody incubation.

**Quantification of immunostaining.** R-loop intensity was quantified using the MetaMorph Microscopy Automation and Image Analysis Software 7.8. For each image, the DAPI signal was used to create a mask of the nucleus in either luminal epithelial compartment or basal/stromal compartments. R-loop intensity was determined by calculating the average intensity in the mask. COBRA1 immunohistochemistry staining images were first separated into haematoxylin and DAB channels using Colour Deconvolution plugin in ImageJ. Intensity of COBRA1 nuclei staining was analysed using the regions of interest (ROIs) and ROI statistics features in Nikon NIS-Elements BR Analysis package. The haematoxylin channel was used to create ROIs in the nucleus of epithelial compartment. The selected ROIs were then pasted to the corresponding DAB channel to determine the mean intensity of COBRA1. At least four images, each of which contained a minimum of one complete epithelial duct, were acquired for each sample. The fields were selected by investigators who were blind to sample genotype. The final intensity for each sample is the average of all images. Quantification of Ki67$^+$ and TUNEL$^+$ cells was done by ImageJ particle analysis software.

**Single-cell isolation of human breast tissue.** Fresh unfixed human breast tissue procured from either cosmetic reduction mammoplasty or mastectomy was processed following the previously published procedure with minor modification[43]. All reagents were purchased from StemCell Technologies (Vancouver, BC, Canada), unless otherwise indicated. Briefly, tissue was minced with scalpels and digested for 15–18 h at 37 °C with shaking in DMEM/F-12 supplemented with 5% FBS, 300 U ml$^{-1}$ collagenase and 100 U ml$^{-1}$ hyaluronidase, 0.1% BSA (Fisher Scientific; BP1600-100), 10 ng ml$^{-1}$ epidermal growth factor (Invitrogen; PHG0311L), 10 ng ml$^{-1}$ cholera toxin (Sigma; C8052), 5 µg ml$^{-1}$ insulin (Sigma; I0526) and 0.5 mg ml$^{-1}$ hydrocortisone (Sigma; H0888). An epithelial-rich organoid was obtained by centrifugation at 100$g$ for 3 min. Red blood cells in the resulting pellets were lysed with 0.8% NH$_4$Cl. Organoids were digested with 0.05% trypsin-EDTA (Life Technologies; 25300), washed with Hank's balanced salt solution supplemented with 2% FBS and then resuspended in 5 mg ml$^{-1}$ Dispase (cat. no. 07913) containing 0.1 mg ml$^{-1}$ DNase I (Roche; 10104159001). Single cells were obtained by going through a 40-mm filter (Fisher; 22363547) to remove remaining cell aggregates.

**Flow cytometry and cell sorting.** Single cells isolated from digested breast tissue were counted and resuspended in Hank's balanced salt solution supplemented with 2% FBS at a concentration of $1 \times 10^6$ cells per 100 µl. Cells were preblocked with 10% rat serum on ice for 10 min, and then stained with a rat anti-CD49f allophycocyanin-conjugated antibody (R&D Systems; clone GOH3) and mouse anti-EpCAM FITC-conjugated antibody (StemCell Technologies; clone VU1-D9) following a previously published protocol[43]. Mouse anti-CD45 (eBiosciences; clone H130), anti-CD235a (eBiosciences; clone HIR2) and anti-CD31 (eBiosciences; clone WM59) biotin-conjugated antibodies were used to label haematopoietic and endothelial cells, followed by Pacific Blue-conjugated Streptavidin (Thermo Fisher Scientific; S11222) incubation. Cells were incubated with 7-aminoactinomycin D (BD Biosciences; 559925) before analysis to distinguish live and dead cells. Cell sorting was performed with a BD FACSAria Flow Cytometer (BD Biosciences). Cells were separated into the following four fractions: EpCAM$^-$CD49f$^-$ stromal cells, EpCAM$^{low}$CD49f$^{high}$ basal epithelial cells, EpCAM$^{high}$ CD49f$^+$ luminal progenitor cells and EpCAM$^{high}$ CD49f$^-$ mature luminal epithelial cells. Purity of

the stromal, basal, luminal progenitor and mature luminal populations were verified by real-time PCR analysis of *VIM* (stromal), *KRT14* (basal), *KRT18* (luminal) and *ESR1* (mature luminal) mRNA using *ACTB* for normalization.

**ChIP assay.** Cells were crosslinked with 1% formaldehyde at room temperature for 10 min, and the reaction was terminated with 125 mM glycine at room temperature for 5 min. The crosslinking reagent was removed by spinning at 1,000$g$ at 4 °C for 5 min, and then cells were washed with cold PBS three times. From this step on to ChIP elution, all buffers were prepared with freshly added cocktail of phosphatase and protease inhibitors (10 mM sodium fluoride, 10 mM sodium pyrophosphate tetrabasic, 2 mM sodium orthovanadate, 1 µg ml$^{-1}$ leupeptin, 1 µg ml$^{-1}$ aprotinin, 1 µg ml$^{-1}$ pepstatin and 1 mM phenylmethylsulfonyl fluoride). Cells were lysed on ice for 10 min using lysis buffer (5 mM HEPES, pH 7.9, 85 mM KCl, 0.5% Triton X-100). Supernatant was removed after spinning at 1,600$g$ at 4 °C for 5 min, and pellet was resuspended with nuclei lysis buffer (50 mM Tris-HCl, pH 8.0, 10 mM EDTA, 1% (wt vol$^{-1}$) SDS). Chromosomal DNA was sonicated using a probe sonicator on ice, then centrifuged at 14,000$g$ for 15 min and the supernatant was saved. Ten per cent of the sonicated DNA was saved as input, the rest was used for ChIP. Antibodies used for ChIP include: anti-Pol II (Abcam; ab817) and anti-NELF-A, which was described previously[44]. Sonicated DNA was incubated with antibody at 4 °C overnight. Dynabeads Protein A (Thermo Fisher Scientific; 10002D) was added the following day and incubated for 4 h. After incubation, Dynabeads was washed two times in TE sarcosyl buffer (50 mM Tris-HCl, pH 8.0, 2 mM EDTA, 0.2% sarcosyl), two times in TSE1 buffer (150 mM sodium chloride, 20 mM Tris-HCl (pH 8.0), 2 mM EDTA, 0.1% (wt vol$^{-1}$) SDS, 1% Triton-X-100), two times in TSE2 buffer (500 mM sodium chloride, 20 mM Tris-HCl, pH 8.0, 2 mM EDTA, 0.1% (wt vol$^{-1}$) SDS, 0.1% Triton X-100), two times in TSE3 buffer (250 mM lithium chloride, 10 mM Tris-HCl, pH 8.0, 1 mM EDTA, 1% sodium deoxycholate, 1% NP-40) and two times in TE buffer (50 mM Tris-HCl, pH 8.0, 2 mM EDTA). Samples were subsequently eluted and reverse-crosslinked and ethanol-precipitated. Locus-specific ChIP signals were assessed by PCR using primers as shown in Supplementary Table 3.

**DNA-RNA immunoprecipitation.** DRIP assay was performed following the established protocol[28]. Briefly, cells were washed two times in PBS and resuspended in TE (Sigma; T9285) containing 5 µl of proteinase K (Roche; 03115828001) and a final concentration of 0.5% SDS. Samples were incubated overnight at 37 °C. Genomic DNA was extracted using phenol–chloroform (Sigma; P2069) in phase lock tubes (5PRIME; 2302840) and ethanol precipitated. DNA was digested using established restriction enzyme cocktail (HindIII, EcoRI, BsrGI, XbaI and SspI) overnight at 37 °C. Digested DNA was cleaned up by phenol–chloroform extraction and ethanol precipitation. For DRIP, digested DNA was incubated with S9.6 antibody overnight at 4 °C in binding buffer (10 mM sodium phosphate, 140 mM sodium chloride, 0.05% Triton X-100 in TE). As a negative control for DRIP, digested DNA was treated overnight with RNase H (NEB; M0297S) and then precipitated for DRIP. Dynabeads were added the next day for 2 h. Bound Dynabeads were then washed with binding buffer three times at room temperature. DNA was eluted, phenol–chloroform extracted and ethanol precipitated. DRIP DNA was sonicated using Covaris (Model S220) before library preparation. Locus-specific DRIP signals were assessed by PCR using primers listed in Supplementary Table 4.

**Library preparation and sequencing.** DRIP-seq libraries were built following the instruction of MicroPlex Library Preparation Kit (Diagenode; C05010011). A total of 15 cycles of PCR amplification was performed. After amplification, libraries were purified using Agencourt AMPure XP System (Beckman Coulter; A63880) following the product manual. Quantity of the libraries was measured with Qubit dsDNA HS Assay Kit (Life Technologies; Q32851), and quality of the libraries was verified using Bioanalyser 2100. Libraries were pooled based on index sequences. Library pool of 14 pM was loaded to Illumina HiSeq2000 and sequenced by 50 bp single-read sequencing module. After sequencing run, demultiplexing with CASAVA was employed to generate the FASTQ file for each sample. Four biological replicates were used for DRIP-seq and between 38 and 114 million total reads were obtained for each biological sample.

**Bioinformatics analysis of DRIP-seq.** Reads in FASTQ file were aligned to human genome by BWA[45], a software package for mapping low-divergent sequences against reference genome, and only unique mapped reads were selected for analysis. We used Perl script to extract the unique mapped reads in four regions, namely TSS region (up- and downstream 2 kb of transcription starting site), TTS region (up- and downstream 2 kb of transcription terminating site), gene body region (downstream 2 kb of TSS to upstream 2 kb of TTS), distal region (upstream 100 to 2 kb of TSS) and gene desert region (1 mb away from any TSS). For each of 31,823 annotated genes with unique TSS and TTS, extracted reads were normalized to total reads of each sample, and averaged for B1 and NC before being applied with a signed $t$-test for differential R-loop in B1 samples. For gene body region that is $<2$ kb in length, a signed rank-sum test was used instead. A log 2FC cutoff 0.8 and adjusted $P$ value 0.05 were used to define differential R-loop for any given gene. $P$ value in Figs 2b and 3a was calculated using permutation test by

randomly regrouping values in NC and B1, to test the significance that average R-loop in B1 is higher than that in NC. MATLAB for Linux was used to carry out the permutation test. Heatmap of reads was generated using MATLAB to show the reads in TSS, TTS, gene body and distal region for genes that are top ranked 5,000 in TSS region. Genes in each subpanel of heatmap are sorted individually.

Peak-calling program MACS (version 1.4, $q$ value $= 0.01$) was used to identify the peaks (R-loop binding sites) from uniquely mapped reads[46]. Peak loci were first extended to nearest enzyme (HindIII, EcoRI, BsrGI, XbaI and SspI), and then TSS-bound peaks were identified by 1 bp overlap to 5′ TSS of human reference genes. For each group of samples (B1 or NC), we selected genes that were associated with TSS-bound peaks in all four samples of that group. Venn diagrams of the overlap genes were generated by BioVenn[47], a web application for comparison and visualization of biological lists. The $P$ value of the significance of the overlap in the Venn diagrams was calculated by Fisher's test.

Pausing index (or travelling ratio) is defined previously[32], which is the ratio of total Pol II density in the promoter-proximal region to the total Pol II density in the transcribed region. Promoter-proximal region is defined as TSS upstream 30 bp to downstream 300 bp, and transcribed region is from TSS downstream 300 bp to TTS. Curves in Fig. 4b,c and Supplementary Fig. 7c are the accumulative distribution of the percentage of Pol II-bound genes with a given pausing index. $P$ values were calculated using signed rank-sum test between the values of NC and B1 at each data point.

HOMER program with default parameter was used for prediction of transcription factor binding sites among TSS regions of genes with *BRCA1* mutation-associated R-loop[48]. The TSS regions chosen as the background include those containing R-loop peaks in both B1 and NC samples and the difference in TSS R-loop intensity hold a $P$ value $> 0.05$.

GC skew[28] was predicted by SkewR, which uses StochHMM, a Hidden Markov Model software, to predict R-loop-binding regions. The program clusters genes into four skew classes as Strong, Weak, No and Reverse Skew. SkewR was trained on hg19 human genome using default parameters. We only used Strong and Weak Skew genes as genes with GC skew.

**Statistics for mouse tumour study/patient sample analysis.** For the clinical sample analysis, treatment groups (carrier, control) were contrasted. For the animal tumour study, time-to-event data were summarized by treatment group with Kaplan–Meier curves and groups were contrasted with regard to time event with log-rank tests. All statistical testing was two sided with a significance level of 5%. SAS Version 9.4 for Windows (SAS Institute, Cary, NC, USA) was used throughout.

**Data availability.** Sequence data that support the findings of this study have been deposited in NIH Gene Expression Omnibus (GEO) with the accession codes GSE96672. All other remaining data are available within the article and Supplementary Files, or available from the authors on request.

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

## Acknowledgements

We thank Dr Anthony Wynshaw-Boris for the MMTV-Cre mouse strain, Dr Stephen Leppla for providing the S9.6 anti-R-loop antibody used in the initial study, the Flow Cytometry Facility and Genome Sequencing Facility at UT Health San Antonio for technical assistance, Dr Exing Wang for assistance in quantification of immunostaining intensity, Drs Luis Palomero and Tianbao Li for bioinformatics assistance, and Dr Ya-ting Hsu for critical reading of the manuscript. The work was supported by grants to Y.H. from the National Institutes of Health (NIH CA170306), to L.-Z.S. from NIH (ES022057), to V.X.J. from NIH (GM114142), to T.J.C from NIH (CA164122) and the Skinner Endowment and the Holly Beach Public Library and to R.L. from NIH (CA161349), the Alamo Breast Cancer Foundation, the Congressionally Directed Medical Research Program (WSlXWH-14-1-0129) and the Tom C. & H. Frost Endowment. H.-C.C. was supported by an NCI T32 Training Grant (T32CA148724). We also thank generous support from Cancer Therapy and Research Center at the University of Texas Health Science Center at San Antonio (P30CA054174). The work was also supported by a grant to Y.H. from the Cancer Prevention Research Institute of Texas CPRIT RP170126.

## Author contributions

R.L. and Y.H. managed the overall project, designed the experiments and wrote the manuscript. X.-W.Z., H.-C.C., Y.W., C.Z., X.-Y.Z. and S.J.N. carried out the experiments, S.S., I.J., M.L., B.O., H.W., A.P., T.S., J.B., F.M., M.A.P., E.P., K.C., C.I., B.N.P., O.O., C.T. and R.E. accrued patients and collected clinical samples. X.-W.Z., H.-C.C., Y.W., L.-Z.S., T.J.C., J.M., F.C., V.X.J., Y.H. and R.L. analysed the data.

## Additional information

**Competing interests:** The authors declare no competing financial interests.

**DOI: 10.1038/ncomms16211**

# Author Correction: Attenuation of RNA polymerase II pausing mitigates BRCA1-associated R-loop accumulation and tumorigenesis

Xiaowen Zhang, Huai-Chin Chiang, Yao Wang, Chi Zhang, Sabrina Smith, Xiayan Zhao, Sreejith J. Nair, Joel Michalek, Ismail Jatoi, Meeghan Lautner, Boyce Oliver, Howard Wang, Anna Petit, Teresa Soler, Joan Brunet, Francesca Mateo, Miguel Angel Pujana, Elizabeth Poggi, Krysta Chaldekas, Claudine Isaacs, Beth N. Peshkin, Oscar Ochoa, Frederic Chedin, Constantine Theoharis, Lu-Zhe Sun, Tyler J. Curiel, Richard Elledge, Victor X. Jin, Yanfen Hu & Rong Li

*Nature Communications* 8:15908 doi: 10.1038/ncomms15908 (2017); Published 26 Jun 2017; Updated 30 Mar 2018

The original version of this Article omitted the following from the Acknowledgements:

'The work was also supported by a grant to Y.H. from the Cancer Prevention Research Institute of Texas CPRIT RP170126.'

This has been corrected in both the PDF and HTML versions of the Article.

