## [Peer Review File · Nature Communications]

Reviewers' comments:

Reviewer #1 (Remarks to the Author):

This study addresses the role of NELF-dependent promoter proximal Pol II pausing in BRCA1 mutation carrier patients. BRCA1 depletion was previously shown to cause R-loop accumulation, implicating BRCA1 in R-loop biology. Here the authors add the Pol II negative elongating factor, NELF into the BRCA1-associated R-loop accumulation and breast cancer development. In particular it is concluded that Pol II pausing contributes to R-loop accumulation in BRCA1 mutation carrier patients and is proposed that BRCA1 is recruited to promoter regions to decrease R-loops caused by Pol II pausing. It is also observed that R-loop accumulation events in BRCA1 mutation patients occur preferentially in the luminal lineage of breast epithelial cells.

BRCA1 is undoubtedly a crucial tumour suppressor and therefore research on how this protein functions is of particular importance. Recently, BRCA1 was shown to associate with R-loop regions and its absence caused R-loop accumulation in these regions. Adding promoter-proximal pausing and NELF in this mechanism is potentially interesting and could provide further insight into both R-loop and BRCA1 connections but also in the mechanism of promoter-proximal pausing itself. This could constitute an important step forward on understanding of the transcription cycle. However I am not convinced that promoter pausing of Pol II in BRCA1 patients occurs in a different manner than in non-carrier individuals and that R-loops indeed accumulate in these individuals due to Pol II pausing, as proposed. Also even though R-loop formation and Pol II promoter-proximal pausing are events dependent on transcription, no nascent transcriptional analysis is shown or how transcription correlates with these findings. I will therefore focus my comments on addressing these above issues.

Major comments:

Figure 1: 1: It is not clear why the ML and LP cells populations exhibit higher R-loop levels. Were they expecting this result? A possible explanation could relate to different levels of transcription in these two cell populations, which is reflected by differences in R-loop levels. A GRO-seq or other nascent RNA analysis comparing levels of nascent transcripts is needed to address this possibility?

2: The differences in R-loop levels between non-carrier and BRCA1-mutation carrier individuals appear to be significant but is that because of the numbers of reads obtained from the DRIP-seq or is it biologically significant? It would be helpful if the authors had included a biological replicate of the DRIP-seq. Also over-expression of RNase H1 is an important control in all DRIP and DRIP-seq experiments to validate the specificity of the R-loop signal and should be included.

3: In figure 1g, where an individual gene is shown, the difference in R-loop signal intensity is not evident for the intron 1 region, where promoter-proximal pausing would be expected to occur. Please comment. For this reason, I think a Pol II ChIP (preferably Ser5P CTD Pol II) should be included to see whether in regions of promoter-proximal pausing the R-loops increase in BRCA1 carriers.

Figure 3: 1. Pausing index of Pol II can be a somewhat vague term and so needs careful definition in these experiments. Specific details and individual examples of how the pausing index is calculated are needed. Rahl et al. 2010 is cited where the pausing index is taken as Pol II promoter proximal density/Pol II gene body density. However in this present study no individual representative genes are shown to illustrate how pausing index is calculated. Is it Ser5P/Ser2P? If so, raw Pol II ChIP data should be shown before the pausing index is calculated. In the Materials and Methods, it is mentioned that Ser2P antibody and Pol II antibody is used; does this mean Ser5P has not been used for pausing calculations? The pausing index is a very crucial point in this study and therefore should be addressed in full detail. Experiments addressing the mechanism are important: how does Pol II Ser5P/Ser2P correlate with increase in R-loops? What happens to this ratio upon RNase H1-over expression?

2. What are the levels of nascent transcription in genes with R-loops and high pausing index? Is it higher? As mentioned above, promoter-proximal pausing of Pol II and R-loops correlate very much with transcription and therefore it is important that nascent transcription analysis is performed and

compared to the data obtained.

3. From Figure 1g, it appears that there is R-loop formation in the non-carrier individuals. Has it been checked how NELF and BRCA1 correlate in these individuals? Or is the NELF-BRCA1 correlation specific for the BRCA1-mutant carriers? If so, why?

4. Does NELFB interact with BRCA1? Has a Co-IP experiment been carried out to be sure that the effects seen in the fig. 3e-3h are direct?

5. The data obtained in figure 3e and 3f are interesting. The concept that R-loops accumulate because of promoter-proximal pausing is intriguing and reasonable. However it is surprising that upon NELFB knock-down, R-loops do not decrease; It would be expected that if R-loops accumulate due to pausing then they would be below WT levels in NELF knock-down. Why is there only an effect when BRCA1 is knocked-down and not in WT conditions? Please comment.

6. It would be interesting to see what the effects of flavopiridol (which should prolong the promoter-proximal pausing) are on R-loop formation in BRCA1 KD conditions.

Figure 4. In figure 4e, I do not think one experiment is sufficient to state 'that reduced tumour incidence in DKO mice is unlikely due to rescue of either HR repair defect or elevated DNA replication stress'

Minor points

1. In the abstract and throughout the text, the authors refer to studies performed in tissue culture cell lines as 'in vitro'. Please rephrase.

2. In the discussion, the authors propose that 'BRCA1 plays an important role in resolving Pol II pausing related R-loop accumulation'. The authors (and other studies) have not shown that BRCA1 can resolve an R-loop as a helicase. Please rephrase.

3. For the imaging figures, the scale should be included in the figures and not only in the figure legends.

Reviewer #2 (Remarks to the Author):

This is a detailed, complex and very logical paper, investigating the function and action of BRCA1 along with other factors in four subcategories of breast cells in relation to breast tumour development. I commend the authors on how they have achieved this

The authors have done a remarkable job of describing such a complex story so briefly, but they have had to use many abbreviations in order to do this. I was not immediately familiar with all these abbreviations and so I have attempted to summarise the findings for myself to check that I have understood correctly.

In addition to BRCA1's known roles in DSB repair and DNA replication stress it has recently been discovered to have a role in the elimination of R-loops.

R-loop intensity (measured by IF) is higher in luminal epithelial cells from BRCA1 mutation carriers than from normals. This pattern isn't obvious in basal and stromal cells from the same patients. This finding has been confirmed using DRIP-seq.

R-loop intensity is most pronounced at the Transcription Start Sites (TSS) of genes, compared to other genomic regions.

The authors have classified 2 groups of genes, which I think are those genes displaying more R-loops in BRCA1 mutation carriers than in normals. The text defining the 2 groups needs to made

easier to understand, as it has defeated me. The authors then describe "using this gene set" but I'm not sure which of the 2 groups they mean, or did they mean both together ..?

The authors have discovered a correlation between genes with TSS R-loops and Pol11 pausing index and they investigate whether these are mechanistically linked, using knockdowns of NELF (Pol11 pausing factor) and BRCA1 in T47D cells: -

Knockdown of BRCA1 increases R-loops in genes with pre-existing R-loop enrichment

Knockdown of NELF, in cells with BRCA1 already knocked down, reduced both R-loops and Pol11 pausing.

In mice, BRCA1 Knock-out luminal cells have high R-loop staining, but double BRCA1 / NELF knock-outs have both reduced R-loop staining and reduced mammary tumourigenesis.

The Summary is written in continuous text, but I wonder if bullet points (if allowed by journal house style) would be easier to follow.

I haven't found a list of abbreviations, but this paper would benefit from one

Reviewer #3 (Remarks to the Author):

Short but interesting paper which I believe would be of interest and value to the DNA damage, transcription and lineage analysis fields once the deficiencies have been addressed by the authors. R-loops are increasingly appreciated as playing important roles in sensitivity to DNA damage in the genome and transcriptional regulation. However, the study of R-loops has been difficult in mammalian cells, in part because the reagents for detecting them are problematic. The paper is for the most part clearly written, but it has some technical/conceptual deficiencies which should be addressed by the authors to raise the strength of the claims being made.

Major difficulties:

1 The central conclusions of the paper rely heavily on DRIP-seq with the S9.6 antibody. There is no validation of the peaks by, for example, qPCR. It would be important to also validate the signals coming from the DRIP-seq by additional methods, such as using RNASE-H Chip to show the signals from S9.6 are coming from R-loops and/or methylation footprinting. Thus no real methodological validation of the veracity of the DRIP-seq results are provided. This is an important deficiency.

2 The observations of reduced tumour burden are difficult to interpret mechanistically. Perhaps the double mutants result in more slowly replicating cells or the replication stress alone could reduce tumour formation without a direct role for BRCA1/NELF as being suggested. Perhaps replication stress alone could explain the reduction in tumours. There is no in depth analysis of the tumours that do form, in relation to the relative rates of cell division and apoptosis in the different genotypes. Understanding which genes are transcriptionally affected in luminal cells in the single and double mutants might help understand whether the DKO are truly reverting their cell state phenotypes.

3 The authors don't explore functionally whether downstream genes that are implicated by involvement with R-loops might plausibly be involved in oncogenesis as well as lineage specification. They note that XBP1 exhibits a pattern similar to that of estrogen stimulated R-loops in vivo. This raises the potential confounding issue of RANK/RANKL signaling in BRCA1 deficiency and the regulation of estrogen signaling through the PgR/ER axis. How do we know that the effects seen here are direct, ie via BRCA1 acting at specific gene loci on R-loop formation, rather than indirect through BRCA1 stimulated paracrine signaling?

4 The choice of cre driver may bias conclusions as to luminal cell lineage effects – MMTV driver is known to be biased towards higher rates of expression/deletion in luminal cell lineages in the mouse and can be quite mosaic in its pattern of deletion. The authors either need to show that target gene deletions are happening equally in basal cell lineages of the mouse mammary tree in the experiments conducted and/or use drivers such as K5 or K14 cre that are more efficient in the basal cell lineages of the mouse mammary gland, to confirm the results are truly lineage related and not a consequence of the cre driver line being chosen.

5 The NELF hemizygous mice behave as wildtype – there is no intermediate phenotype. This doesn't argue strongly for dosage being important in the model as the authors are suggesting.

Minor issues

6 Figures 1f and 2a have dynamite plots representing values that are an average of repeated measures but there is no indication of the variance or standard deviation associated. As such they are uninformative.

7 The NELF clinical outcomes analysis in the supplemental information is meaningless in its current form – the authors do not stratify for effects of known major breast cancer subtypes. Either do a proper multivariable analysis by molecular subgroups or leave this out – it does not add to the paper. I also think it may be over-reaching, this study is of molecular significance in a cellular mechanism, grabbing at clinical correlations is unnecessary.

We thank all three reviewers for their constructive critiques and insightful suggestions that help improve the rigor and overall quality of our work. We have now expanded the manuscript by adding more technical information, experimental data, and a separate discussion section. In addition, the genomic data have been submitted to the NCBI Omnibus database and the accession number is GSE96672. Reviewers have the access to the data through this link: <https://www.ncbi.nlm.nih.gov/geo/query/acc.cgi?token=ynctmgymrpyznat&acc=GSE96672>. Reviewers' concerns are addressed point by point in blue below.

REVIEWER #1:

“This study addresses the role of NELF-dependent promoter proximal Pol II pausing in BRCA1 mutation carrier patients. BRCA1 depletion was previously shown to cause R-loop accumulation, implicating BRCA1 in R-loop biology. Here the authors add the Pol II negative elongating factor, NELF into the BRCA1-associated R-loop accumulation and breast cancer development. In particular it is concluded that Pol II pausing contributes to R-loop accumulation in BRCA1 mutation carrier patients and is proposed that BRCA1 is recruited to promoter regions to decrease R-loops caused by Pol II pausing. It is also observed that R-loop accumulation events in BRCA1 mutation patients occur preferentially in the luminal lineage of breast epithelial cells.”

“BRCA1 is undoubtedly a crucial tumour suppressor and therefore research on how this protein functions is of particular importance. Recently, BRCA1 was shown to associate with R-loop regions and its absence caused R-loop accumulation in these regions. Adding promoter-proximal pausing and NELF in this mechanism is potentially interesting and could provide further insight into both R-loop and BRCA1 connections but also in the mechanism of promoter-proximal pausing itself. This could constitute an important step forward on understanding of the transcription cycle. However I am not convinced that promoter pausing of Pol II in BRCA1 patients occurs in a different manner than in non-carrier individuals and that R-loops indeed accumulate in these individuals due to Pol II pausing, as proposed. Also even though R-loop formation and Pol II promoter-proximal pausing are events dependent on transcription, no nascent transcriptional analysis is shown or how transcription correlates with these findings. I will therefore focus my comments on addressing these above issues.”

To clarify, it was not our intention to conclude that “promoter pausing of Pol II in BRCA1 patients occurs in a different manner than in non-carrier individuals”. Rather, we propose that BRCA1 acts downstream of Pol II pausing to rapidly clear ensuing R-loop accumulation. Thus the difference between BRCA1 mutation carriers and non-carriers lies in the steady-state R-loop levels, *not* Pol II pausing. In further support, BRCA1 depletion in cell culture does not affect Pol II pausing (compare column 1 with 2 in Fig. 5d,e).

Major comments:

“Figure 1: 1: It is not clear why the ML and LP cells populations exhibit higher R-loop levels. Were they expecting this result? A possible explanation could relate to different levels of transcription in these two cell populations, which is reflected by differences in R-loop levels. A GRO-seq or other nascent RNA analysis comparing levels of nascent transcripts is needed to address this possibility?”

We thank the reviewer for raising this excellent question. In support of the explanation offered by the reviewer, a recently published epigenomic and transcriptomic study indicates that normal breast luminal epithelial cells have significantly higher levels of steady-state transcripts (3.7x) and more hypomethylated transcription enhancers (2x) than myoepithelial cells (Gascard et al., *Nature Communications*, 6:6351). In addition, high-resolution, strand-specific DRIPc-seq was recently developed by Dr. Fred Chedin, who is a

co-author in the current manuscript and whose lab pioneered the original DRIP-seq method. Combining DRIPc-seq and PRO-seq in established human cell lines, Dr. Fred Chedin's group found a strong correlation between R-loop formation and nascent transcription (see Fig. S4A, *Molecular Cell*, 63:167). This further corroborates the notion that different R-loop levels are at least partly attributable to different levels of transcription. As another part of the equation, factors that regulate R-loop removal, such as BRCA1, also contribute to the overall R-loop abundance.

We concur with the reviewer that a nascent transcription study using clinical samples would allow us to confirm the published cell culture-based findings. However, we point out that it is exceedingly challenging to acquire sufficient numbers of sorted primary breast epithelial cells for a nuclear run-on reaction. Furthermore, the lengthy procedure of tissue processing, including overnight enzymatic digestion at 37°C and hours-long flow cytometry-based cell sorting preclude us from accurately assessing nascent transcription *in vivo*. We and others in the breast biology field are currently exploring ways to propagate primary epithelial cells *in vitro* while maintaining their lineage-specific identity. If successful, this could allow us to acquire enough primary cells for nascent transcription analysis. In the revised manuscript, we discuss the reviewer's point in the context of the aforementioned technical challenges and publications from other groups.

“2: The differences in R-loop levels between non-carrier and BRCA1-mutation carrier individuals appear to be significant but is that because of the numbers of reads obtained from the DRIP-seq or is it biologically significant? It would be helpful if the authors had included a biological replicate of the DRIP-seq. Also over-expression of RNase H1 is an important control in all DRIP and DRIP-seq experiments to validate the specificity of the R-loop signal and should be included.”

For bioinformatics analysis, extracted reads were normalized to total reads of a given sample. Therefore, the results shown in Figures 2b and 3a represent genuine difference in R-loop levels between the cohorts of BRCA1-mutation carriers and non-carriers. Furthermore, R-loop levels used for statistical analysis are averages of four biological replicates for both cohorts, and the multi-color overlay in the IGV images in Figure 2c represent data from all four biological replicates. These points are now stated clearly in the revised manuscript.

Per reviewer #1's request, we include RNase H pre-treatment as a control in both locus-specific DRIP and genome-wide DRIP-seq. As shown in Supplementary Fig. 4a and 4b, RNase H pre-treatment abolishes R-loop signals in both readouts, thus demonstrating specificity of the R-loop detection assays.

“3: In figure 1g, where an individual gene is shown, the difference in R-loop signal intensity is not evident for the intron 1 region, where promoter-proximal pausing would be expected to occur. Please comment. For this reason, I think a Pol II ChIP (preferably Ser5P CTD Pol II) should be included to see whether in regions of promoter-proximal pausing the R-loops increase in BRCA1 carriers.”

Resolution of the DRIP-seq protocol used in our study is limited by the availability of restriction enzyme sites chosen for genomic DNA digestion. This likely contributes to the relatively broad range of R-loop signals around the promoter-proximal region. Of note, the recently published DRIPc-seq method from Dr. Chedin's group has significantly improved resolution. The clinical samples used in our current DRIP-seq study have been exhausted. Because of the low incidence of BRCA1 mutation carriers in the clinic (2-5% of all breast cancer patients), there is a significant waiting period for acquiring new clinical samples from this group of at-risk women. However, it is our plan to use higher-resolution DRIPc-seq for future R-loop analysis. Whenever sample availability allows, we will also survey promoter-proximal Pol II pausing and related histone modifications using new clinical samples.

“Figure 3: 1. Pausing index of Pol II can be a somewhat vague term and so needs careful definition in these experiments. Specific details and individual examples of how the pausing index is calculated are needed. Rahl et al. 2010 is cited where the pausing index is taken as Pol II promoter proximal density/Pol II gene body density. However in this present study no individual representative genes are shown to illustrate how pausing index is calculated. Is it Ser5P/Ser2P? If so, raw Pol II ChIP data should be shown before the pausing index is calculated. In the Materials and Methods, it is mentioned that Ser2P antibody and Pol II antibody is used; does this mean Ser5P has not been used for pausing calculations? The pausing index is a very crucial point in this study and therefore should be addressed in full detail.”

We apologize for the lack of clarity in the original manuscript. We now follow the previously published paper by Rahl et al (*Cell*, 2010) and use total Pol II signals for Pol II pausing index calculation in both locus-specific ChIP (Figure 5 d-e) and whole-genome ChIP-seq (Figure 4b-c). In the revised manuscript, pausing index for studies is explicitly defined in the text and by illustration in Supplementary Fig. 7a.

“Experiments addressing the mechanism are important: how does Pol II Ser5P/Ser2P correlate with increase in R-loops? What happens to this ratio upon RNase H1-overexpression?”

Per reviewer’s suggestion, we carried out additional correlation analysis of Pol II pausing and R-loops increase. Using total Pol II ChIP-seq data that we generated with human breast tissue, we calculated pausing index and classified genes into two groups: genes with pausing index <10, and genes with pausing index ≥10. We found that genes with higher pausing index have significant more BRCA1-associated R-loop increase (p < 0.01; see figure on the right).

To address the reviewer’s question, we found that RNase H1 overexpression in T47D cells did not lead to significant change in Pol II pausing (Supplementary Fig. 10). Together with other data shown in the manuscript, our findings suggest that, while Pol II pausing leads to elevated R-loop signals, elevated R-loops do not reciprocally influence Pol II pausing.

“2. What are the levels of nascent transcription in genes with R-loops and high pausing index? Is it higher? As mentioned above, promoter-proximal pausing of Pol II and R-loops correlate very much with transcription and therefore it is important that nascent transcription analysis is performed and compared to the data obtained.”

As mentioned above, recently published study from Dr. Chedin’s group clearly shows that R-loop-forming genes have a significantly higher level of nascent transcription downstream of the transcription start site, coinciding precisely with the hotspot of R-loop formation (Figure S4A, *Molecular Cell*, 63, 167). While at present it is technically difficult to conduct nascent transcript analysis of clinical samples, we did find a strong correlation between genes with high Pol II pausing and those with R-loop signals (Fig. 4b), and a significant correlation between *BRCA1* mutation-associated R-loop signals and Pol II pausing in primary breast epithelial cells (Fig. 4c).

To further address the reviewer's comments, we carried out additional analysis of nascent transcripts and pausing index using public data (ChIP-seq from *Nature*, 489:57; GRO-seq from *Molecular Cell*, 58:21). Nascent transcripts were extracted from MCF7 GRO-seq around promoter proximal region (-30bp to +300bp). Pausing index in MCF7 cells was calculated using total Pol II ChIP-seq following methods described in Supplementary Figure 7. Genes were separated into two groups based on pausing index <10 and ≥10. We found that genes with pausing index >10 have significantly higher levels of nascent transcripts (see graph in the previous page). Thus, combined data from cell culture studies and clinical sample analysis support the notion that R-loops and Pol II pausing are associated with high nascent transcription *in vivo*.

“3. From Figure 1g, it appears that there is R-loop formation in the non-carrier individuals. Has it been checked how NELF and BRCA1 correlate in these individuals? Or is the NELF-BRCA1 correlation specific for the BRCA1-mutant carriers? If so, why?”

To address the reviewer's question, we did qPCR-based mRNA analysis of the cancer-free non-carrier samples that we procured. We did not observe any significant correlation between BRCA1 and COBRA1 mRNA levels (data not shown), but this could be due to the limited sample number (n=9). It is known that some triple negative breast cancer (TNBC) tumors that carry WT *BRCA1* gene display the same clinical and molecular features as *BRCA1* mutation-associated tumors (so-called “BRCAness”). Therefore, it is possible that the COBRA1-BRCA1 antagonism could extend beyond *BRCA1*-associated familial breast cancer. We appreciate the reviewer's excellent question, and will test this interesting notion in future studies.

“4. Does NELFB interact with BRCA1? Has a Co-IP experiment been carried out to be sure that the effects seen in the fig. 3e-3h are direct?”

In our earlier publication (*J. Cell Biol.* 155:911), we identified NELFB/COBRA1 through its direct physical association with BRCA1. This protein-protein interaction, which was verified by co-IP and GST pull-down in our published work, likely contributes to the functional antagonism *in vivo* observed in the current study. However, we are cognizant that stringent validation of this notion requires detailed mutational and biochemical characterization, which is beyond the scope of the current study.

“5. The data obtained in figure 3e and 3f are interesting. The concept that R-loops accumulate because of promoter-proximal pausing is intriguing and reasonable. However it is surprising that upon NELFB knock-down, R-loops do not decrease; It would be expected that if R-loops accumulate due to pausing then they would be below WT levels in NELF knock-down. Why is there only an effect when BRCA1 is knocked-down and not in WT conditions? Please comment.”

We thank the reviewer for the insightful question. As discussed in the revised manuscript, we speculate that the R-loop-attenuating activity of BRCA1 is potent enough to rapidly clear R-loops generated by NELF-dependent Pol II pausing in WT BRCA1 cells. This could explain why COBRA1 KD under the WT BRCA1 condition did not lead to further decrease in R-loop signals.

“6. It would be interesting to see what the effects of flavopiridol (which should prolong the promoter-proximal pausing) are on R-loop formation in BRCA1 KD conditions.

In the 2016 *Molecular Cell* paper from Dr. Chedin's group, it was shown that treatment of cells with DRB, another transcription inhibitor with very similar properties to flavopiridol, triggered rapid R-loop turnover (Fig. 1F in *Molecular Cell* 63:167). Per reviewer's suggestion, we conducted a similar experiment

with both DRB and flavopiridol in BRCA1 KD cells. Consistent with the published findings, we found that both treatments significantly reduced R-loop signals (figure on the right). While this may seem paradoxical, one important distinction between NELF-dependent and drug-enhanced Pol II pausing is that the former is a dynamic regulatory process for *active* transcription (e.g., see *Cell* 143, 540 and *J Biol Chem* 285, 6443). In contrast, CDK9 inhibitors such as DRB and flavopiridol “freeze” the transcription apparatus and completely shut off transcription elongation. It is possible that promoter-proximal R-loop formation requires a paused polymerase that at the same time is *poised* for active transcription. While this intriguing observation merits further mechanistic investigation, it really gets to the issue of R-loop dynamics that is outside of the immediate scope of this manuscript.

Figure 4. In figure 4e, I do not think one experiment is sufficient to state that reduced tumour incidence in DKO mice is unlikely due to rescue of either HR repair defect or elevated DNA replication stress¹

In our recently published study (Figure 4, *Nature Communications*, 7:10913), we demonstrated *HR repair-independent* functional antagonism between BRCA1 and COBRA1 by using a GFP reporter assay in cell culture and a Rad51 foci assay *in vivo*. Both assays were used in numerous published studies by multiple laboratories in the HR repair field, including those that establish the HR repair function of BRCA1. In the current manuscript, we used the established 53BP1-staining assay to assess DNA replication stress. Collectively, these findings led us to the conclusion of a DSB repair/replication stress-independent antagonism between *Brcal* and *Cobra1* during tumorigenesis.

Minor points

“1. In the abstract and throughout the text, the authors refer to studies performed in tissue culture cell lines as *in vitro*¹. Please rephrase.”

Done as suggested.

“2. In the discussion, the authors propose that BRCA1 plays an important role in resolving Pol II pausing related R-loop accumulation¹. The authors (and other studies) have not shown that BRCA1 can resolve an R-loop as a helicase. Please rephrase.”

Done as suggested.

“3. For the imaging figures, the scale should be included in the figures and not only in the figure legends.”

Done as suggested.

REVIEWER #2:

“This is a detailed, complex and very logical paper, investigating the function and action of BRCA1 along

with other factors in four subcategories of breast cells in relation to breast tumour development. I commend the authors on how they have achieved this”

We appreciate the reviewer’s positive comments about our work.

“The authors have done a remarkable job of describing such a complex story so briefly, but they have had to use many abbreviations in order to do this. I was not immediately familiar with all these abbreviations and so I have attempted to summarise the findings for myself to check that I have understood correctly.”

“In addition to BRCA1's known roles in DSB repair and DNA replication stress it has recently been discovered to have a role in the elimination of R-loops.”

“R-loop intensity (measured by IF) is higher in luminal epithelial cells from BRCA1 mutation carriers than from normals. This pattern isn't obvious in basal and stromal cells from the same patients. This finding has been confirmed using DRIP-seq.”

“R-loop intensity is most pronounced at the Transcription Start Sites (TSS) of genes, compared to other genomic regions.”

“The authors have classified 2 groups of genes, which I think are those genes displaying more R-loops in BRCA1 mutation carriers than in normals. The text defining the 2 groups needs to be made easier to understand, as it has defeated me. The authors then describe "using this gene set" but I'm not sure which of the 2 groups they mean, or did they mean both together ..?”

We apologize for the confusion. Both groups of genes are used in the gene ontology analysis. This point has been made clear in the revised manuscript.

“The authors have discovered a correlation between genes with TSS R-loops and Pol11 pausing index and they investigate whether these are mechanistically linked, using knockdowns of NELF (Pol11 pausing factor) and BRCA1 in T47D cells:-

Knockdown of BRCA1 increases R-loops in genes with pre-existing R-loop enrichment

Knockdown of NELF, in cells with BRCA1 already knocked down, reduced both R-loops and Pol11 pausing.”

“In mice, BRCA1 Knock-out luminal cells have high R-loop staining, but double BRCA1 / NELF knock-outs have both reduced R-loop staining and reduced mammary tumourgenesis.”

“The Summary is written in continuous text, but I wonder if bullet points (if allowed by journal house style) would be easier to follow.”

“I haven't found a list of abbreviations, but this paper would benefit from one”

Again, we apologize for the dense nature of the original manuscript, as it had originally been written as a Letter for a different *Nature*-series journal, before it was transferred to *Nature Communications*. We have now expanded the text based on the *Nature Communications* format, with more detailed technical information. We also make sure that all abbreviations are spelled out in their first use in the manuscript.

REVIEWER #3:

“Short but interesting paper which I believe would be of interest and value to the DNA damage, transcription and lineage analysis fields once the deficiencies have been addressed by the authors. R-loops are increasingly appreciated as playing important roles in sensitivity to DNA damage in the genome and transcriptional regulation. However, the study of R-loops has been difficult in mammalian cells, in part because the reagents for detecting them are problematic. The paper is for the most part clearly written, but it has some technical/conceptual deficiencies which should be addressed by the authors to raise the

strength of the claims being made.”

Major difficulties:

“1 The central conclusions of the paper rely heavily on DRIP-seq with the S9.6 antibody. There is no validation of the peaks by, for example, qPCR. It would be important to also validate the signals coming from the DRIP-seq by additional methods, such as using RNASE-H Chip to show the signals from S9.6 are coming from R-loops and/or methylation footprinting. Thus no real methodological validation of the veracity of the DRIP-seq results are provided. This is an important deficiency.”

We have addressed the reviewer’s concern by doing the following additional experiments. First, we conducted locus-specific DRIP using a *BRCA1* mutation carrier sample that was pre-treated with RNase H before the IP reaction, and showed that R-loop signals were indeed abolished by RNase H pre-treatment (Supplementary Fig. 4a). Second, we subjected the same RNase H-treated DRIP samples to genome-wide deep-sequencing, and further confirmed the sensitivity of DRIP-seq signals to RNase H pre-treatment (Supplementary Fig. 4b). Third, we were able to validate the elevated R-loop signals in the *BRCA1* mutation carrier samples at a selected number of genomic loci (Supplementary Fig. 3).

“2 The observations of reduced tumour burden are difficult to interpret mechanistically. Perhaps the double mutants result in more slowly replicating cells or the replication stress alone could reduce tumour formation without a direct role for BRCA1/NELF as being suggested. Perhaps replication stress alone could explain the reduction in tumours. There is no in depth analysis of the tumours that do form, in relation to the relative rates of cell division and apoptosis in the different genotypes. Understanding which genes are transcriptionally affected in luminal cells in the single and double mutants might help understand whether the DKO are truly reverting their cell state phenotypes.”

In response to the reviewer’s comments, we conducted histological assessment of tumors from BKO and DKO mice. We did not find any significant difference in tumor cell proliferation (Ki67, Supplementary Fig. 11a) or apoptosis (TUNEL, Supplementary Fig. 11b). One possible explanation could be that the functional antagonism between *BRCA1* and *COBRA1* only influences the early stages of tumor development and thus the incidence of tumor formation. Once breast epithelial cells become tumorigenic, perhaps *COBRA1* depletion would not affect tumor progression in a significant manner.

Per reviewer’s suggestion, we also compared gene expression profiles of BKO and DKO mice, using their corresponding littermate controls as reference. Consistent with previously published studies from other laboratories (e.g., *Oncogene* 30:1597), few transcriptionally active genes in BKO mammary epithelium were affected as compared to their WT control. Likewise, no appreciable transcriptomic differences were observed between BKO and DKO mammary epithelia in either 6 or 8-week old mice (Supplementary Fig. 13 and Table 3). Thus, as discussed in the revised manuscript, gene expression rescue does not appear to account for the differences in R-loop dynamics and tumor incidence between BKO and DKO. Future work will be focused on the mechanistic link between R-loop accumulation and *BRCA1*-associated tumorigenesis.

“3 The authors don’t explore functionally whether downstream genes that are implicated by involvement with R-loops might plausibly be involved in oncogenesis as well as lineage specification. They note that XBP1 exhibits a pattern similar to that of estrogen stimulated R-loops in vivo. This raises the potential confounding issue of RANK/RANKL signaling in BRCA1 deficiency and the regulation of estrogen signaling through the Pgr/ER axis. How do we know that the effects seen here are direct, ie via BRCA1 acting at specific gene loci on R-loop formation, rather than indirect through BRCA1 stimulated paracrine signaling?”

To address the reviewer's comments, we conducted immunohistochemistry of RANKL, ER, and PgR in mammary epithelia of WT, BKO, and DKO mice (Supplementary Fig. 12). While we observed significantly elevated RANKL staining in BKO mice as previously reported, we did not notice any appreciable rescue of RANKL expression in DKO (Supplementary Fig. 12a). No significant changes in PgR/ER expression were observed between BKO and DKO, either (Supplementary Fig. 12b). Thus, the BRCA1/COBRA1 antagonism in R-loop dynamics and tumorigenesis is unlikely simply due to the known paracrine signaling.

“4 The choice of cre driver may bias conclusions as to luminal cell lineage effects - MMTV driver is known to be biased towards higher rates of expression/deletion in luminal cell lineages in the mouse and can be quite mosaic in its pattern of deletion. The authors either need to show that target gene deletions are happening equally in basal cell lineages of the mouse mammary tree in the experiments conducted and/or use drivers such as K5 or K14 cre that are more efficient in the basal cell lineages of the mouse mammary gland, to confirm the results are truly lineage related and not a consequence of the cre driver line being chosen.”

We have confirmed that in our MMTV-Cre KO mutant mice BRCA1 was depleted equally efficiently in luminal and basal cell compartments of the mouse mammary gland (Supplementary Fig. 14). However, the reviewer's question concerning lineage specificity of the BRCA1/COBRA1 antagonism is well taken. In future work, we plan to address this important question by using additional tissue- and cell type-specific Cre systems.

“5 The NELF hemizygous mice behave as wildtype - there is no intermediate phenotype. This doesn't argue strongly for dosage being important in the model as the authors are suggesting.”

We appreciate the reviewer's insightful comment. We concur that the mouse genetic models used in our current study do not recapitulate human cancer development in the most faithful manner. This is also evident in *Brcal* hemizygous mice, which are known not to mimic the elevated breast cancer incidence associated with germ-line *BRCA1* mutation carriers in humans. Thus, there is a need to combine clinical sample studies and preclinical investigation using model systems. This point is made clear in the revised manuscript.

Minor issues

“6 Figures 1f and 2a have dynamite plots representing values that are an average of repeated measures but there is no indication of the variance or standard deviation associated. As such they are uninformative.”

This has been corrected.

“7 The NELF clinical outcomes analysis in the supplemental information is meaningless in its current form - the authors do not stratify for effects of known major breast cancer subtypes. Either do a proper multivariable analysis by molecular subgroups or leave this out - it does not add to the paper. I also think it may be over-reaching, this study is of molecular significance in a cellular mechanism, grabbing at clinical correlations is unnecessary.”

We agree with the reviewer and have removed the supplementary information as suggested.

REVIEWERS' COMMENTS:

Reviewer #1 (Remarks to the Author):

I have read though this revised manuscript as well as the thorough response to the three reviewer's comments. I am impressed by the revisions made to this ms which have addressed my concerns over the original ms as well as tackling the issues raised by my fellow reviewers. Consequently I now feel this paper is worthy of publication by Nature Communications. Clearly the association of BRCA1 mutations in breast cancer derived cells with increased R-loop formation and the interplay of this effect with the NELF subunit COBRA now looks solid and certainly is of significant interest in several associated research fields (transcription and cancer biology)

Reviewer #2 (Remarks to the Author):

I am happy with the changes the authors made in response to my previous comments.

Reviewer #3 (Remarks to the Author):

The revised manuscript has been substantially improved by the additional experiments to support the specificity of the DRIP-seq and also the corroborating experiments on the BKO/DKO mice. The discussion appears to better reflect some of the uncertainties in the mouse model to human comparison. Whether 53BP1 immunoreactivity in the DKO/BKO represents replication stress or breaks in DNA is not totally addressed, however the point about this being increased in the DKO mice is noted.

I have no further major comments.

Response to Reviewers' Comments:

REVIEWERS' COMMENTS:

Reviewer #1 (Remarks to the Author):

I have read though this revised manuscript as well as the thorough response to the three reviewer's comments. I am impressed by the revisions made to this ms which have addressed my concerns over the original ms as well as tackling the issues raised by my fellow reviewers. Consequently I now feel this paper is worthy of publication by Nature Communications. Clearly the association of BRCA1 mutations in breast cancer derived cells with increased R-loop formation and the interplay of this effect with the NELF subunit COBRA now looks solid and certainly is of significant interest in several associated research fields (transcription and cancer biology)

We thank reviewer #1's positive comments.

Reviewer #2 (Remarks to the Author):

I am happy with the changes the authors made in response to my previous comments.

We appreciate reviewer #2's approval.

Reviewer #3 (Remarks to the Author):

The revised manuscript has been substantially improved by the additional experiments to support the specificity of the DRIP-seq and also the corroborating experiments on the BKO/DKO mice. The discussion appears to better reflect some of the uncertainties in the mouse model to human comparison. Whether 53BP1 immunoreactivity in the DKO/BKO represents replication stress or breaks in DNA is not totally addressed, however the point about this being increased in the DKO mice is noted.

I have no further major comments.

We concur with reviewer #3's assessment.